# Block Low-Rank Preconditioner with Shared Basis for Stochastic Optimization

**Jui-Nan Yen**
UCLA
juinanyen@cs.ucla.edu

**Sai Surya Duvvuri**
UT Austin
saisurya@cs.utexas.edu

**Inderjit S. Dhillon**
Google and UT Austin
inderjit@cs.utexas.edu

**Cho-Jui Hsieh**
Google and UCLA
chohsieh@cs.ucla.edu

## Abstract

Adaptive methods with non-diagonal preconditioning have shown state-of-the-art results on various tasks. However, their computational complexity and memory requirement make it challenging to scale these methods to modern neural network architectures. To address this challenge, some previous works have adopted block-diagonal preconditioners. However, the memory cost of storing the block-diagonal matrix remains substantial, leading to the use of smaller block sizes that ultimately leads to suboptimal performance. To reduce the time and memory complexity without sacrificing performance, we propose approximating the diagonal blocks of the second moment matrix by low-rank matrices and enforcing the same basis for the blocks within each layer. We provide theoretical justification for such basis sharing and design an algorithm to efficiently maintain this shared-basis block low-rank approximation during training. Our results on a deep autoencoder and a Transformer benchmark demonstrate that the proposed method outperforms first-order methods with slightly more time and memory usage, while also achieving competitive or superior performance compared to other second-order methods with less time and memory usage.

## 1 Introduction

Diagonal adaptive methods such as Adagrad [10] incorporate an adaptive learning rate schedule for each coordinate of the gradient, which is typically estimated as inverse root mean square of the histories of the coordinate gradient $1/\sqrt{\sum_t (\boldsymbol{g}_t)_i^2}$, for a gradient vector $\boldsymbol{g}_t$. This is equivalent to multiplying the gradient vector with a diagonal matrix, a preconditioner. This approach is adopted in optimizers such as Adam [20], RMSProp [29], which became the preferred approach to training deep neural networks due to their robustness to the learning rate parameter. Several methods later developed preconditioners such as Shampoo [16, 4], GGT [1], which also estimate off-diagonal gradient moments such as $\sum_t (\boldsymbol{g}_t)_i (\boldsymbol{g}_t)_j$ for $i \neq j$ and develop a positive definite preconditioner, while outperforming the diagonal methods in both training and validation performance. However, this approach can have high memory requirements due to additional off-diagonal moment information the optimizer needs to store and the high computational requirement of the inverse operation in computing the preconditioner.

The second-moment matrix $\sum_t \boldsymbol{g}_t \boldsymbol{g}_t^T$ constituting both diagonal and off-diagonal cross-moments is infeasible to store, however, the gradients from deep-neural networks can reveal a structure in this matrix. To simplify this problem we analyze a basic neural network block - linear transformation $\boldsymbol{X} \rightarrow \boldsymbol{XW}$, where $\boldsymbol{X} \in \mathbb{R}^{n \times d_{\text{in}}}$ and $\boldsymbol{W} \in \mathbb{R}^{d_{\text{in}} \times d_{\text{out}}}$ and make two important observations: a) the

37th Conference on Neural Information Processing Systems (NeurIPS 2023).

second-moment matrix $\sum_t \boldsymbol{g}_t \boldsymbol{g}_t^T$, where $\boldsymbol{g}_t \in \mathbb{R}^{d_{\text{in}} d_{\text{out}}}$ is a vectorized form of the gradient with respect to $\boldsymbol{W}$ at iteration $t$, has a block-matrix structure with $d_{\text{out}}$ blocks of size $d_{\text{in}} \times d_{\text{in}}$, with high norm diagonal blocks; b) all the diagonal blocks share similar column and row-space with low dimension. Using a block-diagonal approximation directly can still be infeasible due to $\mathcal{O}(d_{\text{out}} d_{\text{in}}^2)$ memory complexity to store all the blocks. To resolve this, a mini-block approach was used previously in [32], which uses smaller blocks $b < d_{\text{in}}$. However, this approximation can discard important within-layer off-diagonal second-moment information. To alleviate this issue we use observation b) and compute a common low-rank basis $\boldsymbol{B} \in \mathbb{R}^{d_{\text{in}} \times k}$, $k \ll d_{\text{in}}$, and a different residual $\boldsymbol{R}^{(i)} \in \mathbb{R}^{k \times k}$ for each block such that $\{\boldsymbol{B}\boldsymbol{R}^{(1)}\boldsymbol{B}^\top, \ldots, \boldsymbol{B}\boldsymbol{R}^{(d_{\text{out}})}\boldsymbol{B}^\top\}$ approximates the diagonal blocks.

We develop an online algorithm to update the shared basis $\boldsymbol{B}$, for which we derive approximation error guarantees demonstrating the nearness of $\{\boldsymbol{B}\boldsymbol{R}^{(1)}\boldsymbol{B}^\top, \ldots, \boldsymbol{B}\boldsymbol{R}^{(d_{\text{out}})}\boldsymbol{B}^\top\}$ to the original block-diagonal approximation of the gradient second-moment matrix. We also use this to prove a $\mathcal{O}(\sqrt{T})$ regret upper bound in the online convex optimization framework [18, 28], which can translate to convergence guarantees in non-convex stochastic optimization as in [1]. We compare the proposed algorithm with state-of-the-art second-order optimizers along with Adam, which is widely recognized as the most commonly used first-order optimizer. The evaluation is performed on a standard autoencoder benchmark and a Transformer benchmark comprising 19.3 million learnable parameters. Compared with other second-order methods, our algorithm utilizes lower memory and time to perform updates at every iteration while showing better or similar validation and training performance. Specifically, in the Autoencoder benchmark, we show similar training performance as Shampoo while using $3.2\times$ lower time, and $1.5\times$ lower memory. In addition to this, in the Transformer benchmark, we use $3.5\times$ lower time with less memory to obtain $\sim 1.8\%$ improvement in validation error. Further, we are able to achieve better validation errors compared with Adam, even under the same training time budget.

**Notation.** We use boldfaced lower-case letters such as $\boldsymbol{x}$ to denote a vector and boldfaced upper-case letters such as $\boldsymbol{H}$ to denote a matrix. $\boldsymbol{H}_{:,i}$ and $\boldsymbol{H}_{i,:}$ denote the $i$-th column and row of the matrix respectively. $\text{Diag}(\boldsymbol{H}_1, \ldots, \boldsymbol{H}_k)$ denotes a block-diagonal matrix with diagonal blocks $\boldsymbol{H}_1, \ldots, \boldsymbol{H}_k$. $\text{vec}(\boldsymbol{H})$ flattens a matrix into a vector, while $\boldsymbol{A} \otimes \boldsymbol{B}$ denotes the Kronecker product of two matrices. $\|\boldsymbol{H}\|_2, \|\boldsymbol{H}\|_F$ denote the 2-norm and Frobenious norm of a matrix respectively.

## 2 Proposed Method

We consider training a deep neural network that takes an input vector $\boldsymbol{x}$ and produces an output $f(\boldsymbol{\theta}, \boldsymbol{x})$. With a training set $(\boldsymbol{y}_i, \boldsymbol{x}_i)_{i=1}^n$ consisting of $n$ samples and a loss function $\ell$, the network parameters $\boldsymbol{\theta}$ are learned by minimizing the empirical loss $\mathcal{L}$ over the training set:

$$\min_{\boldsymbol{\theta}} \mathcal{L}(\boldsymbol{\theta}) := \frac{1}{n} \sum_{i=1}^n \ell(f(\boldsymbol{\theta}, \boldsymbol{x}_i), \boldsymbol{y}_i).$$

To solve the optimization problem, [10] proposed a full matrix variant of Adagrad, which uses a preconditioner matrix to partially capture the curvature information. Given the stochastic gradient $\boldsymbol{g}_t = \nabla \mathcal{L}_B(\boldsymbol{\theta}_t)$ where $\mathcal{L}_B$ represents the loss of a sampled mini-batch, the preconditioner $\boldsymbol{H}_t$ and the next iterate $\boldsymbol{\theta}_{t+1}$ are computed as follows:

$$\boldsymbol{H}_t = \boldsymbol{H}_{t-1} + \boldsymbol{g}_t \boldsymbol{g}_t^T, \ \ \boldsymbol{\theta}_{t+1} = \boldsymbol{\theta}_t - \alpha_t (\boldsymbol{H}_t^{1/2} + \epsilon \boldsymbol{I})^{-1} \boldsymbol{g}_t, \tag{1}$$

where $\alpha_t$ is the step size and $\epsilon$ is a small constant introduced for numerical stability. As manipulating the full preconditioner matrix $\boldsymbol{H}_t$ is computationally prohibitive, commonly used adaptive optimizers often rely on only the diagonal elements of the preconditioner matrix. Examples include the element-wise version of Adagrad [10] and many of its extensions such as Adam [20].

However, previous studies [16, 23, 1] have shown that in many scenarios, non-diagonal preconditioning can lead to superior results. Hence, it is crucial to design an approximation of the preconditioner that captures more information compared to diagonal Adagrad, while still maintaining practicality for neural network training. This is the primary focus of our paper.

### 2.1 Block-diagonal Structure of the Preconditioner

Due to the substantial size of the full preconditioner matrix in deep neural networks, it is often impractical to employ it directly. Consequently, in line with the majority of existing literature

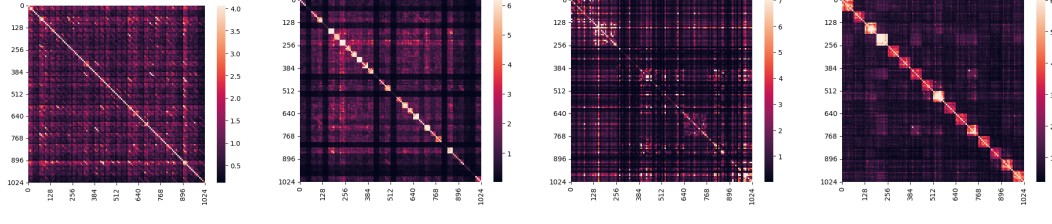

(a) The feed-forward layer with ordering $\boldsymbol{H}_t^{\text{in}}$

(b) The feed-forward layer with ordering $\boldsymbol{H}_t^{\text{out}}$

(c) Self-attention output kernel with ordering $\boldsymbol{H}_t^{\text{in}}$

(d) Self-attention output kernel with ordering $\boldsymbol{H}_t^{\text{out}}$

Figure 1: The second-moment matrices of two different fully connected layers in a Transformer (detailed configurations can be found in our experimental section; details of the visualization can be found in Appendix 6.2). As we can see, $\boldsymbol{H}_t^{\text{out}}$ shows a clearer block diagonal structure compared to $\boldsymbol{H}_t^{\text{in}}$, verifying our choice of grouping weights by output neurons.

[16, 23, 1], we precondition each layer separately. This effectively imposes a block diagonal structure assumption on the preconditioner:

$$\boldsymbol{H}_t = \text{Diag}(\boldsymbol{H}_t^1, \ldots, \boldsymbol{H}_t^L),$$

where $L$ is the number of layers. From now on, we will approximate each $\boldsymbol{H}_t^l$ separately. Therefore, for notational simplicity, we will omit the layer index and refer to the preconditioner matrix for a single layer as $\boldsymbol{H}_t$.

Consider a fully connected layer with weights $\boldsymbol{W} \in \mathbb{R}^{d_{\text{in}} \times d_{\text{out}}}$, where $d_{\text{in}}$ and $d_{\text{out}}$ represent the number of input and output neurons, respectively. In Transformer networks, these dimensions can extend to thousands, making it impractical to store the full preconditioner $\boldsymbol{H}_t$ which has size $d_{\text{in}}d_{\text{out}} \times d_{\text{in}}d_{\text{out}}$. To address this issue, prior studies have identified a block-diagonal structure in $\boldsymbol{H}_t$ and applied a block-diagonal approximation. Specifically, [5, 7] empirically demonstrated that the preconditioner of a fully connected layer exhibits a block-diagonal structure comprising $d_{\text{out}}$ blocks, each with a size of $d_{\text{in}}$. In this structure, the weights connecting different input neurons to the same output neuron are grouped together as a block.

We visualize this block-diagonal structure in Figure 1. Specifically, we examine different fully connected layers of a Transformer model and plot the preconditioner using two different orderings: $\boldsymbol{H}_t^{\text{in}}$ follows row-major ordering, grouping weights associated with the same input neuron, while $\boldsymbol{H}_t^{\text{out}}$ follows column-major ordering, grouping weights associated with the same output neuron. Notably, we observe that $\boldsymbol{H}_t^{\text{out}}$ demonstrates a much stronger block-diagonal structure, confirming our choice of block-diagonal approximation.

Although this particular block-diagonal structure has been observed in the literature, to the best of our knowledge, the underlying reason for this block-diagonal structure has not been analyzed mathematically. Here we show that the block-diagonal structure exists based on the following analysis. Assume we compute the preconditioner with a fixed parameter, then we can denote $\boldsymbol{X} \in \mathbb{R}^{n \times d_{\text{in}}}$ as the input of this layer for all the $n$ samples, and the fully connected layer can be written as

$$\boldsymbol{Z} = \boldsymbol{X}\boldsymbol{W},$$

where $\boldsymbol{W} \in \mathbb{R}^{d_{\text{in}} \times d_{\text{out}}}$ is the weight matrix and $\boldsymbol{Z} \in \mathbb{R}^{n \times d_{\text{out}}}$ is the output matrix. Let $D_{\boldsymbol{W}} = \frac{\partial \mathcal{L}}{\partial \text{vec}(\boldsymbol{W})}$ and $D_{\boldsymbol{Z}} = \frac{\partial \mathcal{L}}{\partial \boldsymbol{Z}}$ denote the gradients of the loss function with respect to $\boldsymbol{W}$ and $\boldsymbol{Z}$, respectively. Additionally, let us define the preconditioner as:

$$\boldsymbol{H} = \begin{bmatrix} \boldsymbol{H}^{(1,1)} & \ldots & \boldsymbol{H}^{(1,m)} \\ \vdots & \ddots & \vdots \\ \boldsymbol{H}^{(m,1)} & \ldots & \boldsymbol{H}^{(m,m)} \end{bmatrix}, \tag{2}$$

where $m = d_{\text{out}}$ is the number of blocks. Since $\boldsymbol{H} = D_{\boldsymbol{W}} D_{\boldsymbol{W}}^T$, we can derive the following lemma:

**Lemma 1** *Assuming the model is evaluated with a fixed parameter $\theta$, then any two diagonal blocks $\boldsymbol{H}^{(i,i)}$, $\boldsymbol{H}^{(j,j)}$ and two corresponding off-diagonal blocks $\boldsymbol{H}^{(i,j)}$, $\boldsymbol{H}^{(j,i)}$ for the preconditioner matrix of a fully connected layer will satisfy the following inequality:*

$$\|\boldsymbol{H}^{(i,i)}\|_F^2 + \|\boldsymbol{H}^{(j,j)}\|_F^2 \geq \|\boldsymbol{H}^{(i,j)}\|_F^2 + \|\boldsymbol{H}^{(j,i)}\|_F^2. \tag{3}$$

The proof is deferred to the appendix. Based on this lemma, we can conclude that the diagonal blocks have a higher Frobenius norm compared with the corresponding off-diagonal blocks.

However, directly using this block-diagonal structure is impractical for large-scale problems. Since there are $d_{\text{out}}$ groups and each group has size $d_{\text{in}}$, storing this block-diagonal approximation requires $O(d_{\text{in}}^2 d_{\text{out}})$ memory. To illustrate, a single fully connected layer in a Transformer with $d_{\text{in}} = 4096$ and $d_{\text{out}} = 1024$ (BERT-large) would occupy over 17GB of memory. To address this challenge, [32] proposed a solution by further dividing each block into mini-blocks. On the other hand, [5] explored an alternative approach that enforces all diagonal blocks to be identical for the larger layers. However, they demonstrated that this method adversely affects performance in practice.

## 2.2 Block Low-rank Approximation with Shared Basis

To simplify notation, we will concentrate on a single layer and omit the layer index moving forward. Let $\boldsymbol{H}_t \approx \text{Diag}(\boldsymbol{H}_t^{(1)}, \dots, \boldsymbol{H}_t^{(m)})$ denotes the block-diagonal approximation of a layer with $m$ blocks with each block containing $b = d/m$ variables. For fully connected layers, we use the grouping mentioned in the previous subsection, so $m = d_{\text{out}}$ and $b = d_{\text{in}}$, while in general, we let $m$ be one of the dimensions for other weight tensors. To reduce the memory requirement, we consider an alternative approach to form low-rank approximations for each diagonal block:

$$\boldsymbol{B}_t^{(i)} \boldsymbol{R}_t^{(i)} (\boldsymbol{B}_t^{(i)})^T \approx \boldsymbol{H}_t^{(i)} = \sum_{j=1}^{t} \boldsymbol{g}_j^{(i)} (\boldsymbol{g}_j^{(i)})^T, \quad \text{for } i = 1, \dots, m,$$

where $k$ is the rank, $\boldsymbol{B}_t^{(i)} \in \mathbb{R}^{b \times k}$ and $\boldsymbol{R}_t^{(i)} \in \mathbb{R}^{k \times k}$ are the eigenvectors and eigenvalues of each block respectively. Using this approach, the memory cost is reduced to $\mathcal{O}(dk)$. Nevertheless, this memory cost is still $k$ times larger than that of first-order optimizers, making it impractical for large-scale training tasks.

We introduce a novel shared-basis approximation scheme to further reduce memory costs. Our key insight is that the diagonal blocks share a similar subspace, which motivates our algorithm to use a joint basis for all the diagonal blocks, thereby reducing memory requirements. To achieve this, we propose the following shared-basis approximation for all the blocks within a single layer:

$$\boldsymbol{H}_t^{(i)} \approx \boldsymbol{B}_t \boldsymbol{R}_t^{(i)} \boldsymbol{B}_t^T, \quad \text{for } i = 1, \dots, m, \tag{4}$$

where $\boldsymbol{B}_t \in \mathbb{R}^{b \times k}$ is the orthogonal shared basis and $\boldsymbol{R}_t^{(i)} \in \mathbb{R}^{k \times k}$ denotes the coefficients for each block. Note that here we assume each $\boldsymbol{R}_t^{(i)}$ is a dense matrix to fully capture the information of each block. This shared-basis approach will reduce the memory complexity to $O(bk + mk^2) = O(d_{\text{in}} k + d_{\text{out}} k^2)$ which can approach the cost of first-order methods when $k^2 \leq d_{\text{in}}$ and $k \leq d_{\text{out}}$. We now give a detailed discussion on this shared-basis block low-rank approximation.

**Why can we share basis?** To support the idea of this shared-basis approximation, we focus on the analysis of fully connected layers and examine the low-rank approximation of each block. Following the analysis presented in Section 2.1, we consider the form of the preconditioner when evaluated on a fixed parameter $\theta$. In such cases, each diagonal block can be expressed as:

$$\boldsymbol{H}^{(i)} = \boldsymbol{X}^T (D_{\boldsymbol{Z}})_{:,i} ((D_{\boldsymbol{Z}})_{:,i})^T \boldsymbol{X}.$$

Therefore, all the diagonal blocks lie in the row space of $\boldsymbol{X}$, the input matrix of a certain layer. Additionally, we observe that in many applications, $\boldsymbol{X}$ tends to occupy a relatively low-rank space. Figure 2 shows the singular value distributions of the intermediate feature $\boldsymbol{X}$ of the Transformer. For all layers, the top $10\%$ of the rank accounts for over $80\%$ of the singular values, which suggests $\boldsymbol{X}$ is low rank. This property has also been observed in the literature [6].

If $\boldsymbol{X}$ can be expressed as low-rank decomposition $\boldsymbol{U \Sigma V}^T$ using SVD, where $\boldsymbol{U} \in \mathbb{R}^{n \times k}, \boldsymbol{\Sigma} \in \mathbb{R}^{k \times k}, \boldsymbol{V} \in \mathbb{R}^{d_{\text{in}} \times k}$, we have

$$\boldsymbol{H}_t^{(i,i)} = \boldsymbol{V \Sigma U}^T (D_{\boldsymbol{Z}})_{:,i} ((D_{\boldsymbol{Z}})_{:,i})^T \boldsymbol{U \Sigma V}^T = \boldsymbol{V C}_i \boldsymbol{V}^T$$

for every $i$, which provides an intuition as to why we can share bases for different blocks. We further verify the performance of our proposed basis-sharing approximation is only slightly worse than forming an independent SVD of each diagonal block. We present these experiments in Appendix 6.3.

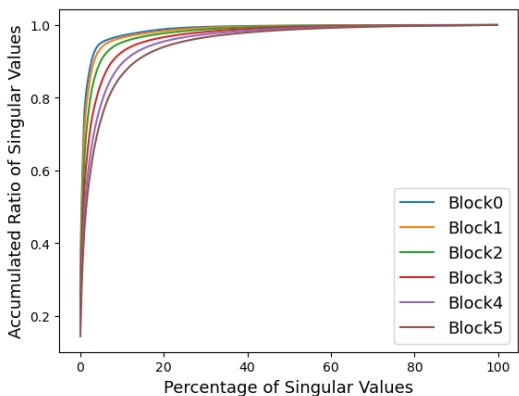

Figure 2: The singular value distribution of the intermediate feature matrix $X$ of 1024 samples after each Transformer block, which verifies the low-rank structure of $X$ in many real-world applications.

**How to compute the shared basis in the online setting?** Given the previous $B_{t-1}, \{R_{t-1}^{(i)}\}_{i=1}^m$ and the current gradient $\{g_t^{(i)}\}_{i=1}^m$, we will describe the procedure for computing the updated approximate matrices $B_t, \{R_t^{(i)}\}_{i=1}^m$. Let

$$M_t^{(i)} \equiv B_{t-1} R_{t-1}^{(i)} (B_{t-1})^T + g_t^{(i)}(g_t^{(i)})^T$$

be the block to be approximated. To derive the share bases, we concatenate all the blocks

$$\bar{M}_t = [M_t^{(1)}, M_t^{(2)}, \ldots, M_t^{(m)}]$$

and compute the SVD to derive its left singular vectors. This is equivalent to computing the top-$k$ eigenvectors of

$$A_t \equiv \bar{M}_t \bar{M}_t^T = \sum\nolimits_{i=1}^m M_t^{(i)} (M_t^{(i)})^T.$$

To efficiently compute the SVD of $A_t$, we adopt the randomized SVD method [17]. This approach enables us to compute the basis without explicitly forming $A_t$ or any of the $M_t^{(i)}$. We initiate the process with an initial matrix $Y \in \mathbb{R}^{b \times (k+p)}$, where $p$ is the over-sampling constant (we set $p = 0$ in experiments). Iteratively, we perform the following update:

$$Y \leftarrow \mathrm{QR}(A_t Y), \tag{5}$$

where $\mathrm{QR}(X)$ denotes the QR decomposition of matrix $X$, returning the Q matrix. Using this approach, $Y$ will converge to the dominant eigenspace of $A_t$. Additionally, since we can initialize $Y$ with the previous basis, only a few iterations are necessary in practice to obtain an accurate estimation.

Further, we have an improved way to compute the matrix product $A_t Y$ in (5). The naive way to calculate $A_t Y = \sum_{i=1}^m M_t^{(i)} (M_t^{(i)} Y)$ takes time complexity of $\mathcal{O}(mbk^2) = \mathcal{O}(dk^2)$. However, we can reduce some repeated computations as the blocks share the same basis. Let

$$p_t^{(i)} \equiv B_{t-1} R_{t-1}^{(i)} (B_{t-1})^T g_t^{(i)},$$

whose time complexity is $\mathcal{O}(mbk) = \mathcal{O}(dk)$. We have

$$\sum\nolimits_{i=1}^m M_t^{(i)} (M_t^{(i)})^T Y = B_{t-1} \left( \sum\nolimits_{i=1}^{d/b} R_{t-1}^{(i)} (R_{t-1}^{(i)})^T \right) (B_{t-1})^T Y + \sum\nolimits_{i=1}^m p_t^{(i)} (g_t^{(i)})^T Y$$
$$+ \sum\nolimits_{i=1}^m g_t^{(i)} (p_t^{(i)})^T Y + \sum\nolimits_{i=1}^m g_t^{(i)} ((g_t^{(i)})^T g_t^{(i)}) (g_t^{(i)})^T Y.$$

Therefore, the time complexity for each power iteration is $\mathcal{O}(m(k^3+bk)+bk^2) = \mathcal{O}(mk^3+dk+bk^2)$ (the QR decomposition takes $\mathcal{O}(bk^2)$ time and is included in this complexity).

**Forming the block low-rank preconditioner with shared basis.** After obtaining the new shared basis $\boldsymbol{B}_t$, we compute the block-dependent $k$-by-$k$ coefficient matrices by

$$\tilde{\boldsymbol{R}}_t^{(i)} = (\boldsymbol{B}_t)^T \boldsymbol{M}_t^{(i)} \boldsymbol{B}_t,$$

which is optimal in terms of minimizing the Frobenius norm reconstruction error for each block. Following the literature of frequent direction approaches [14], we remove the smallest eigenvalue to enable better online updates:

$$\boldsymbol{R}_t^{(i)} = \tilde{\boldsymbol{R}}_t^{(i)} - \sigma_{\min}(\tilde{\boldsymbol{R}}_t^{(i)})\boldsymbol{I}.$$

To account for the missing mass in the low-rank approximation, we track a scalar value for each block using the recurrence relation:

$$\rho_t^{(i)} = \rho_{t-1}^{(i)} + \|\boldsymbol{M}_t^{(i)} - \boldsymbol{B}_t \boldsymbol{R}_t^{(i)} (\boldsymbol{B}_t)^T\|,$$

which can be efficiently computed using power iteration. Finally, to derive the update direction $\boldsymbol{u}_t^{(i)}$, we compute

$$\begin{aligned}
\boldsymbol{u}_t^{(i)} = &- (\boldsymbol{B}_t \boldsymbol{R}_t^{(i)} (\boldsymbol{B}_t)^T + (\rho_t^{(i)} + \epsilon)\boldsymbol{I})^{-1/2} \boldsymbol{g}_t^{(i)} \qquad\qquad (6)\\
= &- \boldsymbol{B}_t (\boldsymbol{R}_t^{(i)} + (\rho_t^{(i)} + \epsilon)\boldsymbol{I})^{-1/2} (\boldsymbol{B}_t)^T \boldsymbol{g}_t^{(i)} + (\rho_t^{(i)} + \epsilon)^{-1/2} \boldsymbol{B}_t (\boldsymbol{B}_t)^T \boldsymbol{g}_t^{(i)} - (\rho_t^{(i)} + \epsilon)^{-1/2} \boldsymbol{g}_t^{(i)}.
\end{aligned}$$

Detailed derivations can be found in Appendix 6.5. Here we follow Shampoo [16] to add the $\epsilon\boldsymbol{I}$ before taking the matrix square root, which is slightly different from the Adagrad formulation shown in (1).

Our overall algorithm is summarized in Algorithm 1. The time complexity of the algorithm is dominated by the matrix products in power iterations, $\mathcal{O}(mk^3 + dk + bk^2)$. The memory complexity is $\mathcal{O}(bk + mk^2)$ for storing the shared basis and the $k \times k$ dense coefficient matrices of each block.

For simplicity, the above discussions are mainly for two-dimensional matrices. For handling higher-order tensor parameters, such as the multi-head self-attention layer in Transformers, we adopt a grouping strategy based on one of the axes. Further discussions can be found in Appendix 6.1.

---

**Algorithm 1** Shared-Basis Low Rank Block-Diagonal Adagrad

---

Initialize: coefficients $\boldsymbol{R}_0^{(i)} = \boldsymbol{0}$, shared bases $\boldsymbol{B}_0 = \boldsymbol{0}$
**for** $t = 1 \ldots T$ **do**
    Compute stochastic gradient $\boldsymbol{g}_t$
    Obtain shared basis $\boldsymbol{B}_t$ by running randomized SVD (Eq. (5)) for $\boldsymbol{A}_t$
    Compute $\tilde{\boldsymbol{R}}_t^{(i)} = (\boldsymbol{B}_t)^T \boldsymbol{M}_t^{(i)} \boldsymbol{B}_t$
    Compute $\boldsymbol{R}_t^{(i)} = \tilde{\boldsymbol{R}}_t^{(i)} - \sigma_{\min}(\tilde{\boldsymbol{R}}_t^{(i)})\boldsymbol{I}$ for $i = 1, \ldots, m$
    Compute $\rho_t^{(i)} = \rho_{t-1}^{(i)} + \|\boldsymbol{M}_t^{(i)} - \boldsymbol{B}_t \boldsymbol{R}_t^{(i)} (\boldsymbol{B}_t)^T\|$ for $i = 1, \ldots, m$
    Compute the preconditioned update $\boldsymbol{u}_t^{(i)} = -(\boldsymbol{B}_t \boldsymbol{R}_t^{(i)} (\boldsymbol{B}_t)^T + (\rho_t^{(i)} + \epsilon)\boldsymbol{I})^{-1/2} \boldsymbol{g}_t^{(i)}$ for $i = 1, \ldots, m$
    Update model parameters $\boldsymbol{\theta}_t = \boldsymbol{\theta}_{t-1} + \alpha_t \boldsymbol{u}_t$
**end for**

---

## 2.3 Regret Bound Analysis

Following previous works [16, 11], we provide a convergence analysis of the proposed algorithm in the online convex optimization framework [18, 28]. In the online convex optimization setting, a parameter $\boldsymbol{\theta}_t \in \mathcal{K}$ is chosen iteratively, where $\mathcal{K}$ is a convex decision set. After the decision of $\theta_t$, a convex loss function $f_t$ is revealed, which can be selected adversarially. The regret suffered by the algorithm up to step $T$ is defined as

$$\text{Regret}_T = \sum_{t=1}^{T} f_t(\theta_t) - \min_{\theta \in \mathcal{K}} \sum_{t=1}^{T} f_t(\theta).$$

We give the following theorem for our proposed method. All the proofs can be found in the Appendix.

**Theorem 1** *Algorithm 1 gives the following regret*

$$Regret_T \leq \sqrt{2}D \sum_{i=1}^{m} \mathrm{Tr}((\boldsymbol{H}_T^{(i)})^{1/2}) + \sqrt{2}b \sum_{i=1}^{m} (\rho_T^{(i)})^{1/2},$$

*where $D$ is the diameter of the constraint set $\mathcal{K}$, and $\boldsymbol{H}_T^{(i)} = \sum_{t=1}^{T} \boldsymbol{g}_t^{(i)}(\boldsymbol{g}_t^{(i)})^T$.*

To bound the second term in Theorem 1, we further derive the following theorem.

**Theorem 2** *Algorithm 1 guarantees*

$$\sum_{i=1}^{m} (\rho_T^{(i)})^{1/2} \leq (3m^3)^{1/4} (\sum_{i=1}^{m} \mathrm{Tr}(\boldsymbol{H}_T^{(i)}))^{1/2},$$

*and thus $Regret_T = \mathcal{O}(\sqrt{T})$.*

This shows that our proposed method is optimal asymptotically in terms of $T$. With some further assumptions, we can further derive bounds of $\rho_T^{(i)}$ in terms of the lower eigenvalues of $\boldsymbol{H}_T^{(i)}$. We defer the results to the Appendix 6.8.

## 2.4 Comparison with Other Methods

We compare the time and space complexity of the proposed algorithm with other second-order methods. We consider the time and memory cost for preconditioning a neural network layer of size $d_{\mathrm{in}} \times d_{\mathrm{out}}$. The results are summarized in Table 1.

As mentioned earlier in Section 2, the full matrix version of Adagrad [10] poses a significant challenge in practice due to the storage and matrix inversion of a full preconditioner with dimensions $d_{\mathrm{in}}^2 \times d_{\mathrm{out}}^2$. To address this issue, previous studies such as [31, 22] proposed low-rank approximation for the preconditioner, which reduces the memory cost to $\mathcal{O}(d_{\mathrm{in}}d_{\mathrm{out}}k)$, with $k$ denoting the rank. However, even with this reduced cost, it remains impractical for modern neural network architectures.

Inspired from the observed empirical block-diagonal structure [7, 26], [32] propose a block diagonal variant of Adagrad and Adam. However, due to the space requirements, storing the block diagonal preconditioner with a block size of $b$ incurs a memory complexity of $\mathcal{O}(db)$. As a result, they are limited to using small block sizes in practice. Similarly, [5] proposes a mini-block Fisher method. For convolutional layers, they group parameters based on filters, resulting in a block size of $r^2$ when the kernel size is $r$. In fully-connected layers, parameter grouping is based on output neurons, yielding a block size of $\mathcal{O}(d_{\mathrm{in}})$ and a memory complexity of $\mathcal{O}(dd_{\mathrm{in}})$. Additionally, they propose maintaining the average of the block diagonal for larger layers, resulting in an approximation of $I \otimes R$, where $R$ represents the average of all diagonal blocks. This approach resembles one side of the Kronecker product methods in the next paragraph, but their research demonstrates some performance loss associated with the approximation.

Another popular approach, such as Shampoo [16] and K-fac [23], involves grouping parameters based on both the input and output neurons, resulting in a Kronecker product approximation $L \otimes R$ for the preconditioner. In order to enhance the time and space complexity of Shampoo, [4] introduces a technique called blocking, which partitions the parameters into smaller blocks. On the other hand, [11] proposes Sketchy as a low-rank variant of Shampoo.

Based on the above discussion, except for Sketchy, all other methods exhibit higher memory complexity compared to our method. Under the same $k$, our memory complexity is larger than Sketchy. This is due to the fact that we are approximating the individual diagonal blocks, but Sketchy only approximates the average of the diagonal blocks (but in both orders). As a result, Sketchy fails to capture heterogeneity between different diagonal blocks. In our experimental evaluation, we demonstrate that when employing the same memory usage (by varying the value of $k$), our proposed method outperforms Sketchy.

## 3 Experimental Results

We evaluate the performance using a standard Autoencoder benchmark [27] on the MNIST dataset [8] and a larger Transformer model [30] on the Universal Dependencies dataset [24].

Table 1: Time and space complexity comparison for a single parameter matrix of size $d_{\text{in}} \times d_{\text{out}}$.

| Algorithm | Time Complexity | Space Complexity |
|---|---|---|
| Full Matrix Adagrad [10] | $d_{\text{in}}^3 d_{\text{out}}^3$ | $d_{\text{in}}^2 d_{\text{out}}^2$ |
| SON-FD [22] | $d_{\text{in}} d_{\text{out}} k^2$ | $d_{\text{in}} d_{\text{out}} k$ |
| Ada-FD [31] | $d_{\text{in}} d_{\text{out}} k^2$ | $d_{\text{in}} d_{\text{out}} k$ |
| Shampoo [16] | $d_{\text{in}}^3 + d_{\text{out}}^3$ | $d_{\text{in}}^2 + d_{\text{out}}^2$ |
| Block Shampoo [4] | $d_{\text{in}} d_{\text{out}} b$ | $d_{\text{in}} d_{\text{out}}$ |
| Sketchy [11] | $d_{\text{in}} d_{\text{out}} k$ | $d_{\text{in}} k + d_{\text{out}} k$ |
| Block Diagonal Adam [32] | $d_{\text{in}} d_{\text{out}} b^2$ | $d_{\text{in}} d_{\text{out}} b$ |
| Ours | $d_{\text{in}} d_{\text{out}} k + d_{\text{in}} k^2 + d_{\text{out}} k^3$ | $d_{\text{in}} k + d_{\text{out}} k^2$ |

We adopt $k = 32$ as the default rank for our methods. For randomized SVD, we set the oversampling parameter to $0$ and the number of iterations to $1$. The impact of these parameters is detailed in our ablation study in Appendix 6.10. Similar to Shampoo, we use the grafting technique [2] in our method. We set the grafting type to `RMSPROP_NORMALIZED`. We compare the proposed optimizer with the widely used Adam optimizer [20] along with the following second-order optimizers:

- Shampoo [16]: we adopt the implementation of [4] and keep most of the default settings with the following differences. Following the setting of [11], for Shampoo, the preconditioning starts at step 101 for stability and the preconditioners are updated every 10 steps for speed (`preconditioning_compute_steps=10`). Furthermore, we explore different block sizes for Shampoo, as suggested by [4]. This approach involves partitioning the parameters into smaller blocks to improve both speed and memory utilization. We denote Shampoo with block size $x$ as Shampoo($x$).

- Sketchy [11]: this is a low-rank approximation version of Shampoo. We follow the setting in their paper to use rank 256 and block size 1024.

- Block Adam [32]: we implement the idea of block-diagonal approximation with a smaller block proposed in [32]. We group the blocks based on output neurons, so we chose the block size to be a divisor of $d_{\text{in}}$. Specifically, a block size of 10 is used for the autoencoder benchmark and a block size of 8 is used for the Transformer benchmark.

We conduct hyperparameter tuning with random hyperparameter search over the search space defined in the appendix. Note that we search over one minus momentum instead of searching momentum directly. Each hyperparameter is sampled from a log-uniform distribution. For the autoencoder benchmark, we conduct 180 trials of random search on one NVIDIA RTX 2080Ti GPU with 11GB memory. For the Transformer benchmark, we conduct 60 trials of random search on one NVIDIA RTX A6000 GPU with 48GB memory.

## 3.1 Autoencoder Benchmark

This autoencoder benchmark has been commonly used to evaluate various optimizers in the literature [23, 3], and we follow the same setting in previous papers. Specifically, we conduct 100 epochs of training using the MNIST dataset. The layer sizes of the autoencoder are $[1000, 500, 250, 30, 250, 500, 1000]$, with a total number of 2.83M learnable parameters. We use tanh as the non-linearity. The batch size is 1000. A linear warmup of 5 epochs is used for learning rate scheduling followed by a linear decay to 0.

In Table 2, we observe that the proposed method has better performance than all the methods except for Shampoo(1000). Compared to Shampoo(1000), our performance is similar while using $3.2\times$ lower time, and $1.5\times$ lower memory. The training loss curve can be seen in Figure 3.

## 3.2 Transformer Benchmark

For the Transformer benchmark, we adopt the part-of-speech tagging task from the Google flax repository [19], which uses the ancient Greek data from the Universal Dependency data sets [24]. The model has 6 Transformer layers. The embedding dimension size is 512 and the MLP dimension size is

Table 2: Experimental results on the autoencoder benchmark.

| Optimizer | Adam | Block Adam | Shampoo(250) | Shampoo(1000) | Sketchy | Ours |
|---|---|---|---|---|---|---|
| Train Loss | 54.66 | 51.78 | 51.56 | **51.22** | 52.41 | 51.49 |
| Time (s) | **50** | 230 | 165 | 553 | 1998 | 140 |
| Memory (MB) | **1593** | 2619 | 2649 | 3641 | 2799 | 1777 |

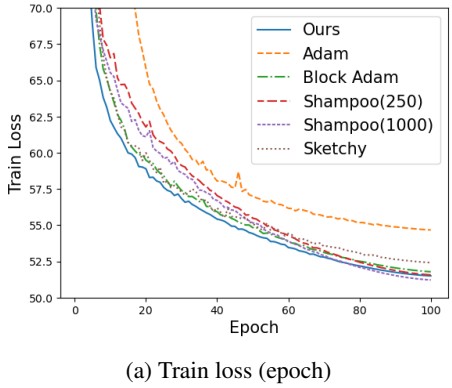

(a) Train loss (epoch)

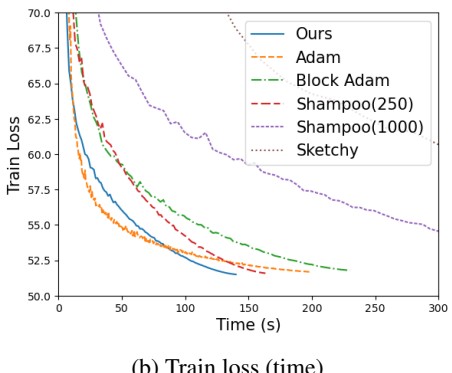

(b) Train loss (time)

Figure 3: Train loss on the autoencoder benchmark. We run Adam for 400 epochs so that it takes roughly the same time as the other methods.

2048. For the self-attention layers, the number of heads is 8, and the query, key, value dimensions are 512. The total number of learnable parameters is 19.3M. We use the default square root learning rate decay scheduling but shorten the training period to 200 epochs and the warmup period to 40 epochs. The batch size is 128. By default, Shampoo excludes the embedding layer from preconditioning as one of its dimensions exceeds 4096. We also follow this setting for a fair comparison, even though our method can be used to precondition the embedding layer due to the reduced memory complexity. We adopt the direction obtained from sgd with momentum for the embedding layer.

In Table 3, we can see that our proposed method performs the best. Compared to Shampoo(1024), we use $3.5\times$ lower time with less memory to obtain $\sim 1.8\%$ improvement in the validation error. The training loss curve and the validation accuracy curve can be seen in Figure 4, where the proposed method outperforms others in terms of both raw training time and number of training steps.

## 4 Related Work

**Second order methods.** For the optimization of ill-conditioned functions, second order methods are often more effective than first order methods. This includes Newton's method and the quasi-Newton methods [9, 21]. Subsampled versions of these methods [15, 25] have also been proposed for solving problems of a larger scale. However, for deep learning problems, the memory requirements associated with these methods become prohibitive. To address this problem, the majority of existing literature preconditions each layer separately. Kronecker product approximation [23, 16] was proposed to reduce the time and space complexity and has shown promising results on several large-scale training tasks. Further, several other approximation schemes have been proposed to exploit second order information, as discussed in Section 2.4.

Table 3: Experimental results on the Transformer benchmark.

| Optimizer | Adam | Block Adam | Shampoo (512) | Shampoo (1024) | Sketchy | Ours |
|---|---|---|---|---|---|---|
| Validation Accuracy | 67.36 | 70.43 | 68.79 | 68.81 | 68.32 | **70.63** |
| Time (min) | **123** | 193 | 325 | 564 | 717 | 161 |
| Memory (MB) | **19157** | 21598 | 19161 | 21209 | 19563 | 19553 |

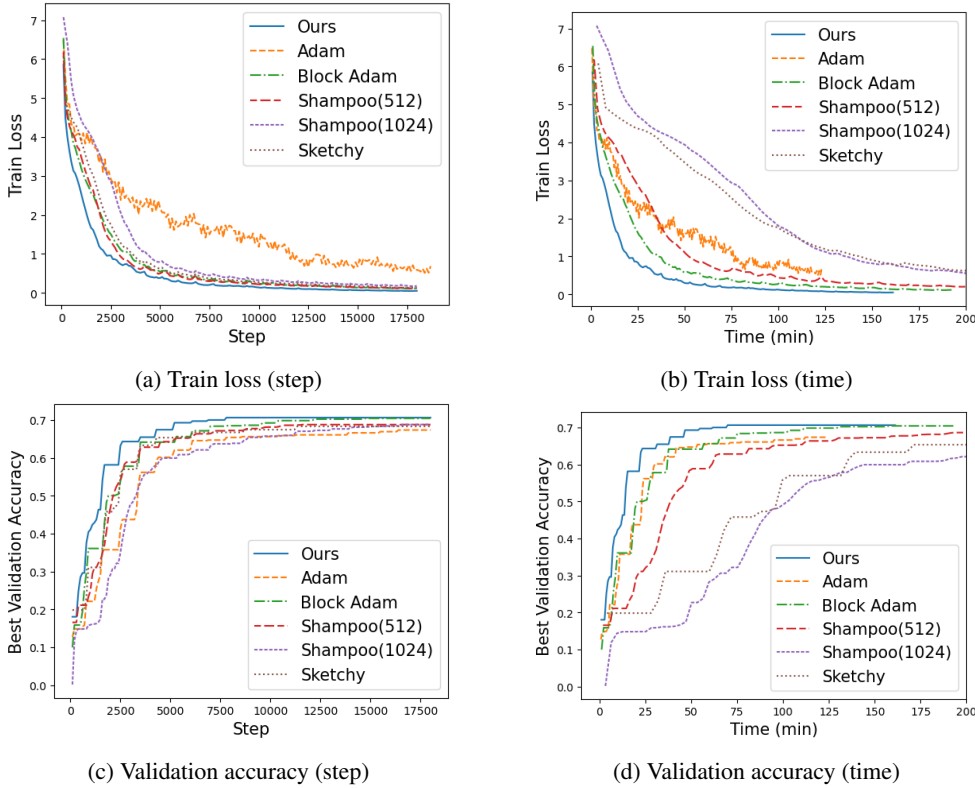

(a) Train loss (step)  (b) Train loss (time)

(c) Validation accuracy (step)  (d) Validation accuracy (time)

Figure 4: Train loss and the best validation accuracy on the Transformer benchmark.

**Frequent direction method.**   Given a stream of vectors, the frequent direction method [13] maintains a low-rank sketch matrix to approximate the covariance matrix of these vectors. The frequent direction method is proved to be optimal in space complexity. This suggests we can approximate the covariance matrix decently even when the vectors are given in a streaming manner. The frequent direction method has been used in previous work for approximating the preconditioner matrix in stochastic optimization [11, 22, 31, 12].

## 5   Conclusions

We have introduced a novel adaptive optimizer that utilizes a shared-basis block low-rank approximation to approximate the full matrix preconditioner. Through evaluations on an autoencoder and a Transformer benchmark, our proposed optimizer has demonstrated superior performance compared to existing first and second-order optimizers, while incurring small memory and time overhead. We also provided convergence guarantees for the proposed optimizer.

## Limitation

The space complexity of our proposed method scales quadratically with $k$, the rank used in the block low-rank approximation. While our empirical results indicate that $k = 32$ is adequate for the Autoencoder and Transformer benchmarks examined in our study, this may still pose a burden when training large-scale models. Exploring methods to reduce the dependence on $k$ while preserving performance will be an interesting future direction.

## Acknowledgements

We thank the anonymous reviewers for their helpful comments. JNY and CJH are supported in part by NSF under 2008173, 2048280, 2325121 and 2331966, Cisco and Sony.

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
