# 6  Appendix

## 6.1  Block-diagonal Structure of Transformers

We observe that for the self-attention layers, the correlation of weights for the same head is stronger. Additionally, the best grouping might depend on the type of the layer (e.g., key, query, value, or output kernel).

To simplify the implementation, we treat all the different kernels in the self-attention as a type of fully-connected layer. For the key, query, and value kernels, the head dimension can be viewed as a part of the output dimension, while for the output kernel, the head dimension can be viewed as a part of the input dimension. After combining the head dimension with either the input or the output dimension, we use the $\boldsymbol{H}_t^{\text{out}}$ ordering as what is used for the fully-connected layers.

## 6.2  Visualization of Figure 1

We down-sample along each dimension to make the computation feasible. To relate with the Frobenius norm, we compute the square of each element and normalize the value. We also cap the top 1% value and average over an 8x8 block to give better visualization.

## 6.3  Approximation Error of Shared vs Non-shared Low-Rank Approximation

We compare the approximation error of the following: Approximate $\boldsymbol{H}_T^{(i)}$ directly with shared and non-shared basis, approximate $\boldsymbol{H}_T^{(i)}$ with frequent direction where $\boldsymbol{g}_t^{(i)}$ is given one by one. We vary the rank $k$ and also compare them with the block diagonal approximation of block size $k$.

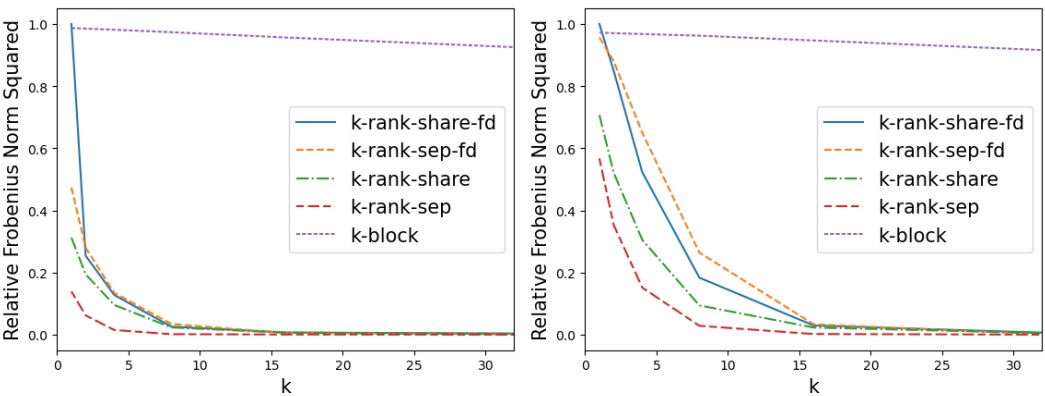

(a) The key kernel of the self-attention layer.    (b) The out kernel of the self-attention layer.

Figure 5: The approximation error comparison for different approximation methods.

In Figure 5, we show the approximation error comparison for different approximation methods. We compute the relative Frobenius norm squared

$$\frac{\sum_{i=1}^{m} \|\boldsymbol{H}_T^{(i)} - \boldsymbol{X}\|_F^2}{\sum_{i=1}^{m} \|\boldsymbol{H}_T^{(i)}\|_F^2},$$

where $\boldsymbol{X}$ is the approximation to be compared.

From the figures, we can see that basis-sharing approximation is only slightly worse than forming an independent SVD of each diagonal block. Additionally, we can see that even though $\boldsymbol{g}_t^{(i)}$ is given one by one, the approximation error is similar to the case where $\boldsymbol{H}_T^{(i)}$ can be accessed directly. Lastly, we can see that the approximation error of our method is much lower than the block diagonal approximation of the same $k$, even though our algorithm has lower time and space complexity.

## 6.4 Proof of Lemma 1

From $\boldsymbol{Z} = \boldsymbol{X}\boldsymbol{W}$, we have
$$D_{\boldsymbol{W}} = \boldsymbol{X}^T D_{\boldsymbol{Z}}.$$

Due to $\|\boldsymbol{A}\|_F^2 = \mathrm{Tr}(\boldsymbol{A}\boldsymbol{A}^T)$ and the cyclic property of trace, we have

$$\|\boldsymbol{H}^{(i,j)}\|_F^2 = \|(D_{\boldsymbol{W}})_{:,i}(D_{\boldsymbol{W}})_{:,j}^T\|_F^2 = \mathrm{Tr}((D_{\boldsymbol{W}})_{:,i}(D_{\boldsymbol{W}})_{:,j}^T(D_{\boldsymbol{W}})_{:,j}(D_{\boldsymbol{W}})_{:,i}^T)$$
$$= \mathrm{Tr}((D_{\boldsymbol{W}})_{:,i}^T(D_{\boldsymbol{W}})_{:,i}(D_{\boldsymbol{W}})_{:,j}^T(D_{\boldsymbol{W}})_{:,j}) = \|\boldsymbol{X}^T(D_{\boldsymbol{Z}})_{:,i}\|_2^2\|\boldsymbol{X}^T(D_{\boldsymbol{Z}})_{:,j}\|_2^2.$$

Consequently, we have

$$\|\boldsymbol{H}^{(i,i)}\|_F^2 + \|\boldsymbol{H}^{(j,j)}\|_F^2 = \|\boldsymbol{X}^T(D_{\boldsymbol{Z}})_{:,i}\|_2^4 + \|\boldsymbol{X}^T(D_{\boldsymbol{Z}})_{:,j}\|_2^4$$
$$\geq 2\|\boldsymbol{X}^T(D_{\boldsymbol{Z}})_{:,i}\|_2^2\|\boldsymbol{X}^T(D_{\boldsymbol{Z}})_{:,j}\|_2^2 = \|\boldsymbol{H}^{(i,j)}\|_F^2 + \|\boldsymbol{H}^{(j,i)}\|_F^2,$$

which completes the proof.

## 6.5 Derivation of Equation 6

For simplicity, let $c = (\rho_t^{(i)} + \epsilon)$. Since $B_t^T B_t = I$, we have

$$B_t^T(I - B_t B_t^T) = B_t^T - B_t^T = 0$$

and

$$(I - B_t B_t^T)B_t = B_t - B_t = 0.$$

Consequently, we have

$$[B_t(R_t^{(i)} + cI)^{-1/2}B_t^T + c^{-1/2}(I - B_t B_t^T)]^2 = [B_t(R_t^{(i)} + cI)^{-1}B_t^T + c^{-1}(I - B_t B_t^T)],$$

as the other two terms are 0.

Similarly, we have

$$[B_t(R_t^{(i)}+cI)^{-1}B_t^T+c^{-1}(I-B_t B_t^T)][B_t(R_t^{(i)}+cI)B_t^T+c(I-B_t B_t^T)] = B_t B_t^T+(I-B_t B_t^T) = I.$$

Combining the two equations proves that

$$[B_t(R_t^{(i)}+cI)^{-1/2}B_t^T+c^{-1/2}(I-B_t B_t^T)] = [B_t(R_t^{(i)}+cI)B_t^T+c(I-B_t B_t^T)]^{-1/2} = [B_t(R_t^{(i)})B_t^T+cI]^{-1/2}.$$

## 6.6 Proof of Theorem 1

To prove Theorem 1, we first prove the following lemmas.

Let
$$\boldsymbol{H}_t^{(i)} = \sum_{j=1}^t \boldsymbol{g}_j^{(i)}(\boldsymbol{g}_j^{(i)})^T$$

be the preconditioner used by block diagonal Adagrad,

$$\hat{\boldsymbol{H}}_t^{(i)} = \boldsymbol{B}_t \boldsymbol{R}_t^{(i)}(\boldsymbol{B}_t)^T + \rho_t^{(i)}\boldsymbol{I}$$

be the preconditioner used by our proposed method, we have

**Lemma 2**
$$\boldsymbol{H}_t^{(i)} \preccurlyeq \hat{\boldsymbol{H}}_t^{(i)} \preccurlyeq \boldsymbol{H}_t^{(i)} + 2\rho_t^{(i)}\boldsymbol{I},$$

*which implies*

$$(\boldsymbol{H}_t^{(i)})^{1/2} \preccurlyeq (\hat{\boldsymbol{H}}_t^{(i)})^{1/2} \preccurlyeq (\boldsymbol{H}_t^{(i)} + 2\rho_t^{(i)}\boldsymbol{I})^{1/2}$$

*and*

$$(\boldsymbol{H}_t^{(i)})^{-1/2} \succcurlyeq (\tilde{\boldsymbol{H}}_t^{(i)})^{-1/2} \succcurlyeq (\boldsymbol{H}_t^{(i)} + 2\rho_t^{(i)}\boldsymbol{I})^{-1/2}.$$

**Proof of Lemma 2**

We have
$$B_t R_t^{(i)} B_t^T + (\rho_t^{(i)} - \rho_{t-1}^{(i)})I = B_t R_t^{(i)} B_t^T + \|M_t^{(i)} - B_t R_t^{(i)}(B_t)^T\|I \succcurlyeq M_t^{(i)}.$$

Adding $\rho_{t-1}^{(i)}I$ on both sides, we get
$$\hat{H}_t^{(i)} = B_t R_t^{(i)} B_t^T + \rho_t^{(i)}I \succcurlyeq M_t^{(i)} + \rho_{t-1}^{(i)}I$$
$$= B_{t-1} R_{t-1}^{(i)}(B_{t-1})^T + g_t^{(i)}(g_t^{(i)})^T + \rho_{t-1}^{(i)}I = \hat{H}_{t-1}^{(i)} + g_t^{(i)}(g_t^{(i)})^T.$$

Consequently,
$$\hat{H}_t^{(i)} = \sum_{j=1}^{t} \hat{H}_j^{(i)} - \hat{H}_{j-1}^{(i)} \succcurlyeq \sum_{j=1}^{t} g_j^{(i)}(g_j^{(i)})^T = H_t^{(i)}. \tag{7}$$

On the other hand, we also have
$$B_t R_t^{(i)} B_t^T \preccurlyeq M_t^{(i)} + \|M_t^{(i)} - B_t R_t^{(i)}(B_t)^T\|I = M_t^{(i)} + (\rho_t^{(i)} - \rho_{t-1}^{(i)})I.$$

Adding $\rho_t^{(i)}I$ on both sides, we get
$$\hat{H}_t^{(i)} = B_t R_t^{(i)} B_t^T + \rho_t^{(i)}I \preccurlyeq M_t^{(i)} + (2\rho_t^{(i)} - \rho_{t-1}^{(i)})I$$
$$= B_{t-1} R_{t-1}^{(i)}(B_{t-1})^T + g_t^{(i)}(g_t^{(i)})^T + \rho_{t-1}^{(i)}I + (2\rho_t^{(i)} - 2\rho_{t-1}^{(i)})I$$
$$= \hat{H}_{t-1}^{(i)} + g_t^{(i)}(g_t^{(i)})^T + (2\rho_t^{(i)} - 2\rho_{t-1}^{(i)})I.$$

Consequently,
$$\hat{H}_t^{(i)} = \sum_{j=1}^{t} \hat{H}_j^{(i)} - \hat{H}_{j-1}^{(i)} \preccurlyeq \sum_{j=1}^{t} g_j^{(i)}(g_j^{(i)})^T + (2\rho_t^{(i)} - 2\rho_{t-1}^{(i)})I = H_t^{(i)} + 2\rho_t^{(i)}I. \tag{8}$$

Combining (7) and (8) completes our proof.

We also utilize the following property of the block diagonal Adagrad preconditioner.

**Lemma 3 (Lemma 5.13, 5.14 [18])** *For the block diagonal Adagrad preconditioner $H_t^{(i)}$, we have*
$$\sum_{t=1}^{T} (g_t^{(i)})^T (H_t^{(i)})^{-1/2} g_t^{(i)} \leq 2\operatorname{Tr}((H_T^{(i)})^{-1/2}).$$

Lastly, we utilize the following lemma for online convex optimization.

**Lemma 4 (Lemma 5.13 [18])** *For online convex optimization, if $\theta_t$ is updated as $\theta_{t+1} = \theta_t - \eta X_t g_t$, then we have*
$$Regret_T \leq \frac{1}{2\eta}\|\theta_1 - \theta_*\|_{X_1^{-1}}^2 + \frac{\eta}{2}\sum_{t=1}^{T}(g_t)^T X_t g_t + \frac{1}{2\eta}\sum_{t=2}^{T}(\theta_t - \theta_*)^T(X_t^{-1} - X_{t-1}^{-1})(\theta_t - \theta_*).$$

**Proof of Theorem 1**

In our proposed method, $\theta_t$ is updated as
$$\theta_{t+1}^{(i)} = \theta_t^{(i)} - \eta(\hat{H}_t^{(i)})^{-1/2} g_t^{(i)}.$$

Thus, by Lemma 4, we have
$$\text{Regret}_T \leq \frac{1}{2\eta}\sum_{i=1}^{m}\|\theta_1^{(i)} - \theta_*^{(i)}\|_{(\hat{H}_1^{(i)})^{1/2}}^2 + \frac{\eta}{2}\sum_{i=1}^{m}\sum_{t=1}^{T}(g_t^{(i)})^T(\hat{H}_t^{(i)})^{-1/2}g_t^{(i)}$$
$$+ \frac{1}{2\eta}\sum_{i=1}^{m}\sum_{t=2}^{T}(\theta_t^{(i)} - \theta_*^{(i)})^T((\hat{H}_t^{(i)})^{1/2} - (\hat{H}_{t-1}^{(i)})^{1/2})(\theta_t^{(i)} - \theta_*^{(i)}). \tag{9}$$

For the first term, by Lemma 2, we have

$$
\|\theta_1^{(i)} - \theta_*^{(i)}\|^2_{(\hat{\boldsymbol{H}}_1^{(i)})^{1/2}} \leq D^2 \operatorname{Tr}((\hat{\boldsymbol{H}}_1^{(i)})^{1/2}) \leq D^2(\operatorname{Tr}((\boldsymbol{H}_1^{(i)} + 2\rho_1^{(i)}\boldsymbol{I})^{1/2}))
$$
$$
\leq D^2(\operatorname{Tr}((\boldsymbol{H}_1^{(i)})^{1/2}) + \sqrt{2}b(\rho_1^{(i)})^{1/2}) \leq (\sqrt{2}b+1)D^2G, \tag{10}
$$

where G is the upper bound of the gradient norm.

For the second term, by Lemma 2 and Lemma 3, we have

$$
\sum_{t=1}^{T}(\boldsymbol{g}_t^{(i)})^T(\hat{\boldsymbol{H}}_t^{(i)})^{-1/2}\boldsymbol{g}_t^{(i)} \leq \sum_{t=1}^{T}(\boldsymbol{g}_t^{(i)})^T(\boldsymbol{H}_t^{(i)})^{-1/2}\boldsymbol{g}_t^{(i)} \leq 2\operatorname{Tr}((\boldsymbol{H}_T^{(i)})^{1/2}) \leq 2\operatorname{Tr}((\hat{\boldsymbol{H}}_T^{(i)})^{1/2}). \tag{11}
$$

For the third term, by Lemma 2, we have

$$
\sum_{t=2}^{T}(\theta_t^{(i)} - \theta_*^{(i)})^T((\hat{\boldsymbol{H}}_t^{(i)})^{1/2} - (\hat{\boldsymbol{H}}_{t-1}^{(i)})^{1/2})(\theta_t^{(i)} - \theta_*^{(i)})
$$
$$
\leq \sum_{t=1}^{T} D^2\|(\hat{\boldsymbol{H}}_t^{(i)})^{1/2} - (\hat{\boldsymbol{H}}_{t-1}^{(i)})^{1/2}\| \tag{12}
$$
$$
\leq \sum_{t=1}^{T} D^2 \operatorname{Tr}((\hat{\boldsymbol{H}}_t^{(i)})^{1/2} - (\hat{\boldsymbol{H}}_{t-1}^{(i)})^{1/2}) = D^2 \operatorname{Tr}((\hat{\boldsymbol{H}}_T^{(i)})^{1/2}),
$$

where the second inequality follows the fact that the trace of a positive definite matrix is larger than its spectral norm.

Furthermore,

$$
\operatorname{Tr}((\hat{\boldsymbol{H}}_T^{(i)})^{1/2}) \leq \operatorname{Tr}((\boldsymbol{H}_T^{(i)} + 2\rho_t^{(i)}\boldsymbol{I})^{1/2})
$$
$$
\leq \operatorname{Tr}((\boldsymbol{H}_t^{(i)})^{1/2}) + \operatorname{Tr}((2\rho_t^{(i)}\boldsymbol{I})^{1/2}) = \operatorname{Tr}((\boldsymbol{H}_t^{(i)})^{1/2}) + \sqrt{2}b(\rho_t^{(i)})^{1/2}. \tag{13}
$$

Combining (9), (10), (11), (12), and (13), and setting $\eta = D/\sqrt{2}$, we get

$$
\operatorname{Regret}_T \leq (\sqrt{2}b+1)D^2G + \sqrt{2}D\sum_{i=1}^{m}\operatorname{Tr}((\hat{\boldsymbol{H}}_T^{(i)})^{1/2})
$$
$$
\leq (\sqrt{2}b+1)D^2G + \sqrt{2}D\sum_{i=1}^{m}(\operatorname{Tr}((\boldsymbol{H}_t^{(i)})^{1/2}) + \sqrt{2}b(\rho_t^{(i)})^{1/2}),
$$

which completes the proof.

### 6.7 Proof of Theorem 2

By triangle inequality, we have

$$
\rho_T^{(i)} = \sum_{t=1}^{T}\|\boldsymbol{M}_t^{(i)} - \boldsymbol{B}_t\boldsymbol{R}_t^{(i)}\boldsymbol{B}_t^T\| \leq \sum_{t=1}^{T}\|\boldsymbol{B}_t\tilde{\boldsymbol{R}}_t^{(i)}\boldsymbol{B}_t^T - \boldsymbol{B}_t\boldsymbol{R}_t^{(i)}\boldsymbol{B}_t^T\| + \|\boldsymbol{M}_t^{(i)} - \boldsymbol{B}_t\tilde{\boldsymbol{R}}_t^{(i)}\boldsymbol{B}_t^T\|. \tag{14}
$$

For the first term, since $\tilde{\boldsymbol{R}}_t^{(i)} \succcurlyeq \boldsymbol{R}_t^{(i)} \succcurlyeq \boldsymbol{0}$, we have

$$
\sum_{t=1}^{T}\|\boldsymbol{B}_t\tilde{\boldsymbol{R}}_t^{(i)}\boldsymbol{B}_t - \boldsymbol{B}_t\boldsymbol{R}_t^{(i)}\boldsymbol{B}_t\|_2 \leq \sum_{t=1}^{T}\operatorname{Tr}(\boldsymbol{B}_t\tilde{\boldsymbol{R}}_t^{(i)}\boldsymbol{B}_t - \boldsymbol{B}_t\boldsymbol{R}_t^{(i)}\boldsymbol{B}_t)
$$
$$
\leq \sum_{t=1}^{T}\operatorname{Tr}(\boldsymbol{M}_t^{(i)} - \boldsymbol{B}_t\boldsymbol{R}_t^{(i)}\boldsymbol{B}_t) = \sum_{t=1}^{T}\operatorname{Tr}(\boldsymbol{g}_t^{(i)}(\boldsymbol{g}_t^{(i)})^T) = \operatorname{Tr}(\boldsymbol{H}_T^{(i)}). \tag{15}
$$

For the second term, since the Frobenius norm or a matrix is larger than its spectral norm, we have

$$\sum_{i=1}^{m} \|\boldsymbol{M}_t^{(i)} - \boldsymbol{B}_t \tilde{\boldsymbol{R}}_t^{(i)} \boldsymbol{B}_t\|_F^2 \geq \sum_{i=1}^{m} \|\boldsymbol{M}_t^{(i)} - \boldsymbol{B}_t \tilde{\boldsymbol{R}}_t^{(i)} \boldsymbol{B}_t\|_2^2.$$

We use

$$\mathrm{stack}(\{\boldsymbol{A}^{(i)}\}_{i=1}^m) \in \mathbb{R}^{m \times b \times b}$$

to denote a stack of matrices. Since $\boldsymbol{B}_t$ is the top-$k$ left singular vectors of

$$\bar{\boldsymbol{M}}_t = [\boldsymbol{M}_t^{(1)}, \boldsymbol{M}_t^{(2)}, \ldots, \boldsymbol{M}_t^{(m)}],$$

by definition, $\mathrm{stack}(\{\boldsymbol{B}_t \tilde{\boldsymbol{R}}_t^{(i)} \boldsymbol{B}_t\}_{i=1}^m)$ is equal to the HOSVD of $\mathrm{stack}(\{\boldsymbol{M}_t^{(i)}\}_{i=1}^m)$. From the quasi-optimality of HOSVD, we have

$$\sum_{i=1}^{m} \|\boldsymbol{M}_t^{(i)} - \boldsymbol{B}_t \tilde{\boldsymbol{R}}_t^{(i)} \boldsymbol{B}_t\|_F^2 = \|\mathrm{stack}(\{\boldsymbol{M}_t^{(i)}\}_{i=1}^m) - \mathrm{stack}(\{\boldsymbol{B}_t \tilde{\boldsymbol{R}}_t^{(i)} \boldsymbol{B}_t\}_{i=1}^m)\|_F^2$$

$$\leq 3\|\mathrm{stack}(\{\boldsymbol{M}_t^{(i)}\}_{i=1}^m) - \mathrm{stack}(\{\boldsymbol{B}_{t-1} \tilde{\boldsymbol{R}}_{t-1}^{(i)} \boldsymbol{B}_{t-1}\}_{i=1}^m)\|_F^2$$

$$= 3\sum_{i=1}^{m} \|\boldsymbol{M}_t^{(i)} - \boldsymbol{B}_{t-1} \tilde{\boldsymbol{R}}_{t-1}^{(i)} \boldsymbol{B}_{t-1}\|_F^2 = 3\sum_{i=1}^{m} \|\boldsymbol{g}_t^{(i)} (\boldsymbol{g}_t^{(i)})^T\|_F^2 = 3\sum_{i=1}^{m} \|\boldsymbol{g}_t^{(i)}\|^4.$$

Consequently, we have

$$\sum_{i=1}^{m} \|\boldsymbol{M}_t^{(i)} - \boldsymbol{B}_t \tilde{\boldsymbol{R}}_t^{(i)} \boldsymbol{B}_t\| \leq m\Big(\sum_{i=1}^{m} \|\boldsymbol{M}_t^{(i)} - \boldsymbol{B}_t \tilde{\boldsymbol{R}}_t^{(i)} \boldsymbol{B}_t\|_2^2/m\Big)^{1/2}$$

$$\leq m\Big(\sum_{i=1}^{m} \|\boldsymbol{M}_t^{(i)} - \boldsymbol{B}_t \tilde{\boldsymbol{R}}_t^{(i)} \boldsymbol{B}_t\|_F^2/m\Big)^{1/2} \leq m\Big(3\sum_{i=1}^{m} \|\boldsymbol{g}_t^{(i)}\|^4/m\Big)^{1/2} \leq m(3/m)^{1/2} \sum_{i=1}^{m} \|\boldsymbol{g}_t^{(i)}\|^2, \tag{16}$$

where the first inequality comes from the concavity of the square root function.

Therefore,

$$\sum_{t=1}^{T} \sum_{i=1}^{m} \|\boldsymbol{M}_t^{(i)} - \boldsymbol{B}_t \tilde{\boldsymbol{R}}_t^{(i)} \boldsymbol{B}_t\| \leq m(3/m)^{1/2} \sum_{i=1}^{m} \sum_{t=1}^{T} \|\boldsymbol{g}_t^{(i)}\|^2 = m(3/m)^{1/2} \sum_{i=1}^{m} \mathrm{Tr}(\boldsymbol{H}_T^{(i)}).$$

This gives

$$\sum_{i=1}^{m} \Big(\sum_{t=1}^{T} \|\boldsymbol{M}_t^{(i)} - \boldsymbol{B}_t \tilde{\boldsymbol{R}}_t^{(i)} \boldsymbol{B}_t\|\Big)^{1/2} \leq m\Big(\sum_{t=1}^{T} \sum_{i=1}^{m} \|\boldsymbol{M}_t^{(i)} - \boldsymbol{B}_t \tilde{\boldsymbol{R}}_t^{(i)} \boldsymbol{B}_t\|/m\Big)^{1/2}$$

$$= m\Big((3/m)^{1/2} \sum_{i=1}^{m} \mathrm{Tr}(\boldsymbol{H}_T^{(i)})\Big)^{1/2} = (3m^3)^{1/4} \Big(\sum_{i=1}^{m} \mathrm{Tr}(\boldsymbol{H}_T^{(i)})\Big)^{1/2}. \tag{17}$$

Finally, from (14), (15), and (17), we have

$$\sum_{i=1}^{m} (\rho_T^{(i)})^{1/2} \leq \sum_{i=1}^{m} \Big(\Big(\sum_{t=1}^{T} \|\boldsymbol{B}_t \tilde{\boldsymbol{R}}_t^{(i)} \boldsymbol{B}_t^T - \boldsymbol{B}_t \boldsymbol{R}_t^{(i)} \boldsymbol{B}_t^T\|\Big)^{1/2} + \Big(\sum_{t=1}^{T} \|\boldsymbol{M}_t^{(i)} - \boldsymbol{B}_t \tilde{\boldsymbol{R}}_t^{(i)} \boldsymbol{B}_t^T\|\Big)^{1/2}\Big)$$

$$\leq \sum_{i=1}^{m} \mathrm{Tr}(\boldsymbol{H}_T^{(i)})^{1/2} + (3m^3)^{1/4} \Big(\sum_{i=1}^{m} \mathrm{Tr}(\boldsymbol{H}_T^{(i)})\Big)^{1/2}, \tag{18}$$

which completes the proof.

## 6.8 Bounding Escaped Mass by the Lower Eigenvalues

The escaped mass $\rho_T^{(i)}$ can be divided into two parts.

$$\rho_T^{(i)} = \sum_{t=1}^{T} \|\boldsymbol{M}_t^{(i)} - \boldsymbol{B}_t \boldsymbol{R}_t^{(i)} \boldsymbol{B}_t^T\| \leq \sum_{t=1}^{T} \|\boldsymbol{B}_t \tilde{\boldsymbol{R}}_t^{(i)} \boldsymbol{B}_t^T - \boldsymbol{B}_t \boldsymbol{R}_t^{(i)} \boldsymbol{B}_t^T\| + \|\boldsymbol{M}_t^{(i)} - \boldsymbol{B}_t \tilde{\boldsymbol{R}}_t^{(i)} \boldsymbol{B}_t^T\|. \quad (19)$$

The first part is in the original frequent direction process, while the second part is introduced because we are sharing the basis.

We define the quality indicator at each step $t$ as the following.

$$Q_T^{(i)} = 1 - \frac{\sum_{t=1}^{T} \|\boldsymbol{M}_t^{(i)} - \boldsymbol{B}_t \tilde{\boldsymbol{R}}_t^{(i)} \boldsymbol{B}_t^T\|}{\sum_{t=1}^{T} \|\boldsymbol{B}_t \tilde{\boldsymbol{R}}_t^{(i)} \boldsymbol{B}_t^T - \boldsymbol{B}_t \boldsymbol{R}_t^{(i)} \boldsymbol{B}_t^T\| + \|\boldsymbol{M}_t^{(i)} - \boldsymbol{B}_t \tilde{\boldsymbol{R}}_t^{(i)} \boldsymbol{B}_t^T\|} \quad (20)$$

This can be interpreted as the complement of the ratio of the approximation error we additionally introduced. We always have $0 \leq Q_T^{(i)} \leq 1$, and we have $Q_T^{(i)} = 1$ when the top-k eigenvectors are the same for all the blocks for every time step.

Based on $Q_T^{(i)}$, we have the following bound.

**Theorem 3** *Algorithm 1 guarantees that for any $i$ and $p < kQ_T^{(i)}$, we have*

$$(\rho_T^{(i)})^{1/2} \leq \left( \frac{1}{kQ_T^{(i)} - p} \sum_{j=p+1}^{b} \sigma_j(\boldsymbol{H}_T^{(i)}) \right)^{1/2},$$

*where $\sigma_j$ denotes the $j$th largest eigenvalues.*

**Proof of Theorem 3**

By definition, we have

$$\boldsymbol{H}_T^{(i)} - \boldsymbol{B}_T \boldsymbol{R}_T^{(i)} \boldsymbol{B}_T^T = \sum_{t=1}^{T} (\boldsymbol{g}_t^{(i)} (\boldsymbol{g}_t^{(i)})^T - \boldsymbol{B}_t \boldsymbol{R}_t^{(i)} \boldsymbol{B}_t^T + \boldsymbol{B}_{t-1} \boldsymbol{R}_{t-1}^{(i)} \boldsymbol{B}_{t-1}^T) = \sum_{t=1}^{T} (\boldsymbol{M}_t^{(i)} - \boldsymbol{B}_t \boldsymbol{R}_t^{(i)} \boldsymbol{B}_t^T). \quad (21)$$

From (21) and the definition of $\tilde{\boldsymbol{R}}_t^{(i)}$ and $\boldsymbol{R}_t^{(i)}$, we have

$$\begin{aligned}
&\mathrm{Tr}(\boldsymbol{H}_T^{(i)}) - \mathrm{Tr}(\boldsymbol{B}_T \boldsymbol{R}_T^{(i)} \boldsymbol{B}_T^T) \\
&= \sum_{t=1}^{T} \mathrm{Tr}(\boldsymbol{M}_t^{(i)} - \boldsymbol{B}_t \boldsymbol{R}_t^{(i)} \boldsymbol{B}_t^T) \geq \sum_{t=1}^{T} \mathrm{Tr}(\boldsymbol{B}_t \tilde{\boldsymbol{R}}_t^{(i)} \boldsymbol{B}_t^T - \boldsymbol{B}_t \boldsymbol{R}_t^{(i)} \boldsymbol{B}_t^T) \\
&= \sum_{t=1}^{T} k \|\boldsymbol{B}_t \tilde{\boldsymbol{R}}_t^{(i)} \boldsymbol{B}_t^T - \boldsymbol{B}_t \boldsymbol{R}_t^{(i)} \boldsymbol{B}_t^T\|
\end{aligned} \quad (22)$$

From (19) and (20), we have

$$Q_T^{(i)} \rho_T^{(i)} \leq \sum_{t=1}^{T} \|\boldsymbol{B}_t \tilde{\boldsymbol{R}}_t^{(i)} \boldsymbol{B}_t^T - \boldsymbol{B}_t \boldsymbol{R}_t^{(i)} \boldsymbol{B}_t^T\| \quad (23)$$

Combining (22) and (23), we get

$$\mathrm{Tr}(\boldsymbol{H}_T^{(i)}) - \mathrm{Tr}(\boldsymbol{B}_T \boldsymbol{R}_T^{(i)} \boldsymbol{B}_T^T) \geq kQ_T^{(i)} \rho_T^{(i)}. \quad (24)$$

On the other hand, let $\boldsymbol{v}_1, \ldots, \boldsymbol{v}_b$ be the eigenvectors of $\boldsymbol{H}_T^{(i)}$, from (21), we have

$$
\begin{aligned}
&\operatorname{Tr}(\boldsymbol{H}_T^{(i)}) - \operatorname{Tr}(\boldsymbol{B}_T \boldsymbol{R}_T^{(i)} \boldsymbol{B}_T^T) \\
&= \sum_{j=1}^b \boldsymbol{v}_j \boldsymbol{H}_T^{(i)} \boldsymbol{v}_j^T - \sum_{j=1}^b \boldsymbol{v}_j \boldsymbol{B}_T \boldsymbol{R}_T^{(i)} \boldsymbol{B}_T^T \boldsymbol{v}_j^T \leq \sum_{j=1}^b \boldsymbol{v}_j \boldsymbol{H}_T^{(i)} \boldsymbol{v}_j^T - \sum_{j=1}^p \boldsymbol{v}_j \boldsymbol{B}_T \boldsymbol{R}_T^{(i)} \boldsymbol{B}_T^T \boldsymbol{v}_j^T \\
&= \sum_{j=1}^p \boldsymbol{v}_j (\boldsymbol{H}_T^{(i)} - \boldsymbol{B}_T \boldsymbol{R}_T^{(i)} \boldsymbol{B}_T^T) \boldsymbol{v}_j^T + \sum_{j=p+1}^b \boldsymbol{v}_j \boldsymbol{H}_T^{(i)} \boldsymbol{v}_j^T \\
&= \sum_{j=1}^p \boldsymbol{v}_j (\boldsymbol{H}_T^{(i)} - \boldsymbol{B}_T \boldsymbol{R}_T^{(i)} \boldsymbol{B}_T^T) \boldsymbol{v}_j^T + \sum_{j=p+1}^b \sigma_j(\boldsymbol{H}_T^{(i)}) \\
&= \sum_{j=1}^p \sum_{t=1}^T \boldsymbol{v}_j (\boldsymbol{M}_t^{(i)} - \boldsymbol{B}_t \boldsymbol{R}_t^{(i)} \boldsymbol{B}_t^T) \boldsymbol{v}_j^T + \sum_{j=p+1}^b \sigma_j(\boldsymbol{H}_T^{(i)}) \\
&\leq p \sum_{t=1}^T \| \boldsymbol{M}_t^{(i)} - \boldsymbol{B}_t \boldsymbol{R}_t^{(i)} \boldsymbol{B}_t^T \| + \sum_{j=p+1}^b \sigma_j(\boldsymbol{H}_T^{(i)}) \\
&= p \rho_T^{(i)} + \sum_{j=p+1}^b \sigma_j(\boldsymbol{H}_T^{(i)})
\end{aligned}
\tag{25}
$$

Combining (24) and (25), we have

$$
k Q_T^{(i)} \rho_T^{(i)} \leq \operatorname{Tr}(\boldsymbol{H}_T^{(i)}) - \operatorname{Tr}(\boldsymbol{B}_T \boldsymbol{R}_T^{(i)} \boldsymbol{B}_T^T) \leq p \rho_T^{(i)} + \sum_{j=p+1}^b \sigma_j(\boldsymbol{H}_T^{(i)}).
$$

Consequently, for any $p < k Q_T^{(i)}$, we have

$$
\rho_T^{(i)} \leq \frac{1}{k Q_T^{(i)} - p} \sum_{j=p+1}^b \sigma_j(\boldsymbol{H}_T^{(i)}),
$$

and thus

$$
(\rho_T^{(i)})^{1/2} \leq \left( \frac{1}{k Q_T^{(i)} - p} \sum_{j=p+1}^b \sigma_j(\boldsymbol{H}_T^{(i)}) \right)^{1/2},
$$

which completes the proof.

## 6.9 Hyperparmaeter Search Space

We list the search space for hyperparameters in Table 4. Similar to Adam, we also use the second moment parameter $\beta_2$ for our proposed method. For the autoencoder benchmark, we set the weight decay to 0.

Table 4: The search space for hyperparameters.

| Hyperparameter | Range |
|---|---|
| Learning rate | $[10^{-4}, 10^{-2}]$ |
| Weight decay | $[10^{-3}, 10^{-1}]$ |
| Momentum $1 - \beta_1$ | $[10^{-4}, 10^{-1}]$ |
| Second moment $1 - \beta_2$ | $[10^{-4}, 10^{-1}]$ |

## 6.10 Hyperparmaeter Sensitivity

To understand the effect of the various parameters of our methods. We conduct an ablation study on the number of iterations, the rank, and the oversampling parameters used by randomized SVD. We also experiment with the effect of initializing $\Omega$ using that from the last round for randomized SVD.

From Table 5 and 6, we can see that increasing the rank improves the performance of our proposed method. We can also see that our proposed method is still competitive with Adam with ranks smaller than 32.

Table 5: Comparison of our proposed method with different ranks on the autoencoder benchmark.

| Optimizer | Adam | Rank-1 | Rank-2 | Rank-4 | Rank-8 | Rank-16 | Rank-32 |
|---|---|---|---|---|---|---|---|
| Train Loss | 54.66 | 55.29 | 53.51 | 52.88 | 52.42 | 51.94 | **51.49** |

Table 6: Comparison of our proposed method with different ranks on the transformer benchmark. We use the best parameters obtained from Rank-32 and take the average result of 10 runs.

| Optimizer | Adam | BlockAdam | Rank-1 | Rank-4 | Rank-8 | Rank-16 | Rank-32 |
|---|---|---|---|---|---|---|---|
| Validation Accuracy | 67.70 | 70.36 | 70.30 | 70.41 | 70.35 | 70.38 | **70.51** |
| Time (min) | **123** | 193 | 140 | 143 | 153 | 149 | 161 |
| Memory (MB) | **19157** | 21598 | 19550 | 19553 | 19553 | 19553 | 19553 |

For the rest of the ablation study, we focus on the autoencoder benchmark. Figure 6 shows the corresponding curves for the train loss. From Table 7, we can see that reusing $\Omega$ from the last round improves the performance and the benefit is larger when the rank is smaller.

Table 7: Comparison of our proposed method regarding whether to initialize $\Omega$ using the value from the last round.

| Optimizer | Rank-16 (no init) | Rank-16 | Rank-24 (no init) | Rank-24 | Rank-32 (no init) | Rank-32 |
|---|---|---|---|---|---|---|
| Train Loss | 52.09 | 51.94 | 51.71 | 51.67 | 51.51 | **51.49** |
| Time (s) | **94** | **94** | 126 | 128 | 142 | 140 |
| Memory (MB) | 1777 | 1777 | 1779 | 1779 | 1777 | 1777 |

From Table 8 and Table 9, we find that increasing the number of iterations and the oversampling parameter does not necessarily improve the performance. Thus, we set them to 1 and 0 respectively in our main experiments.

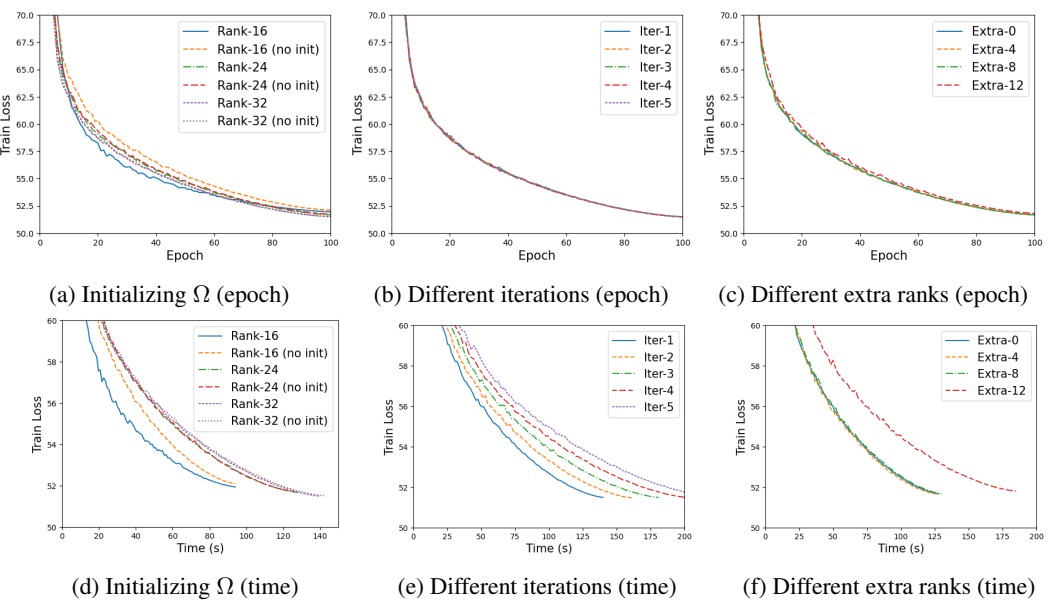

| | | | | |
| (a) Initializing $\Omega$ (epoch) | | (b) Different iterations (epoch) | | (c) Different extra ranks (epoch) |
| (d) Initializing $\Omega$ (time) | | (e) Different iterations (time) | | (f) Different extra ranks (time) |

Figure 6: Train loss for ablation study on autoencoder.

Table 8: Comparison of our proposed method with rank set to 32, oversampling parameter set to 0, and the number of randomized SVD iterations set to different values.

| Optimizer | Iter-1 | Iter-2 | Iter-3 | Iter-4 | Iter-5 |
|---|---|---|---|---|---|
| Train Loss | 51.49 | **51.47** | 51.49 | 51.48 | 51.48 |
| Time (s) | **140** | 161 | 181 | 203 | 224 |
| Memory (MB) | **1777** | 1779 | 1779 | 1779 | 1781 |

Table 9: Comparison of our proposed method with rank set to 24 and different oversampling parameters.

| Optimizer | Extra-0 | Extra-4 | Extra-8 | Extra-12 |
|---|---|---|---|---|
| Train Loss | 51.67 | **51.66** | 51.67 | 51.80 |
| Time (s) | 128 | **126** | 130 | 185 |
| Memory (MB) | 1779 | 1779 | 1779 | 1779 |

## 6.11 Additional Experiments

To verify the effectiveness of our method, we conduct the following additional experiments. We mainly focus on the comparison with Adam.

### 6.11.1 Autoencoder Benchmark on Cifar-10

We use the same autoencoder architecture and learning rate schedule as in Section 3.1 to conduct 100 epochs of training on the Cifar-10 dataset.

Table 10 shows that similar to the MNIST data set, our proposed method performs better than Adam and Block Adam.

### 6.11.2 ResNet Experiment on Cifar-10

We conduct image classification experiments on Cifar-10 with ResNet20 (0.27M parameters) and ResNet56 (0.85M parameters). We train for 100k and 200k steps with a batch size of 128. A linear

Table 10: Experimental results on the autoencoder benchmark on Cifar-10.

| Optimizer | Adam | Block Adam | Ours (rank 32) |
|---|---|---|---|
| Train Loss | 1776.20 | 1764.41 | 1758.05 |

warmup of 5 epochs is used for learning rate scheduling followed by a cosine decay to 0. We conduct 60 trials of random hyperparameter search for each setting.

Table 11: Image classification results on Cifar-10 with ResNet20.

| Optimizer | Adam (100k step) | Ours (100k step) | Adam (200k step) | Ours (200k step) |
|---|---|---|---|---|
| Test Accuracy | 92.07 | 92.35 | 92.79 | 93.02 |

Table 12: Image classification results on Cifar-10 with ResNet56.

| Optimizer | Adam (100k step) | Ours (100k step) | Adam (200k step) | Ours (200k step) |
|---|---|---|---|---|
| Test Accuracy | 92.22 | 93.04 | 94.02 | 94.40 |

Table 11 and 12 show that our method performs better than Adam on both ResNet20 and ResNet56. The performance gap naturally decreases as the number of steps increases as both methods should reach similar performance after running for a sufficient amount of steps.

### 6.11.3 Experiment on LLM

We conduct a small-scale experiment for LLM on one of the smaller gpt2-models (7.3M parameters). We train on the WikiText-103 dataset (over 100M tokens) for 10k steps with a batch size of 128. A linear warmup of 1k steps is used for learning rate scheduling followed by a cosine decay to 0.1 of the peak learning rate. For the embedding layer, we adopt the direction obtained from Adam.

Table 13 shows the experiment result. Our method is over 25% faster in steps and 15% faster in time to reach the same level of validation perplexity as Adam.

Table 13: Small-scale LLM experiment on WikiText-103.

| Optimizer | Adam | Ours (rank 32) |
|---|---|---|
| Validation Perplexity | 55.92 | 54.33 |
| Time (min) | 121 | 130 |