# OpenReview forum: "Block Low-Rank Preconditioner with Shared Basis for Stochastic Optimization"
_NeurIPS.cc/2023/Conference — NeurIPS 2023 poster_

### Official Review · Reviewer_BzFF · 2023-07-05

**Soundness:** 4 excellent
**Presentation:** 3 good
**Contribution:** 3 good
**Rating:** 6
**Confidence:** 3

**Summary:**

In this paper, the authors introduce a novel method to perform adaptive gradient updates. Similarly to other works in the literature, this method focuses on capturing curvature information by considering an approximation to the second-moment matrix $H_t = \sum_t g_t g_t^T$, where $g_t$ is the gradient at the t-th optimisation step, and uses this as a preconditioner to apply to the gradient direction before performing the next optimisation step. Given the size of the target matrix $H_t$, using it in its entirety is unfeasible: much of the literature then focuses on slimming its structure by proposing suitable block-diagonal approximations of $H_t$. This work builds upon these approaches.

The proposed preconditioner is studied for fully-connected layers, and performs the following approximations:
- Drop all inter-layer dependencies in $H_t$ (that is, consider a block-diagonal approximation where each block $H_t^l$ refers to the parameters in a single layer $l$ - this is already considered in the literature)
- Further sub-block-diagonalise each of the $H_t^l$: the blocks are picked so to group together parameters affecting the same output variable
- Consider a low-rank approximation of these sub-blocks. Crucially, the approximation is made on the same manifold for each sub-block within a given $H_t^l$
This approach is justified by the following observations:
- The norm of each pair of diagonal sub-blocks is larger than that of their off-diagonals counterparts (Figure 1 and lemma 1, justifying sub-block-diagonalisation)
- Each diagonal sub-block referring to the same layer lies on the row-space of the input matrix of the given layer, and this is generally close to low-rank (justifying sharing of low-rank basis)

The paper then proceeds to analyse the performance of the novel method, both theoretically (providing memory and convergence bounds) and empirically (comparing its performance against available alternatives, mostly in terms of memory usage and time-to-solution). The tests (conducted on two commonly used benchmarks, considering fully-connected layers in a transformer and an auto-encoder) confirm the validity of the method.

**Strengths:**

- The approach is supported and justified by a (reasonably solid) theoretical analysis, which greatly contributes to grounding the intuition behind it
- The method seems effective, and the experiments conducted are convincing enough in validating its performance - at least compared to other methods which try to approximate $H_t$ in a similar fashion
- The paper is well-structured, and reads cleanly (especially after some minor proofreading - see comments below)

**Weaknesses:**

- The approach might be rather limited in its applicability, as it mostly focuses on fully-connected layers: it is unclear to me if/how easily this analysis could be extended to other types of layers (say convolutional, for example). I invite the authors to further comment on this
- The work might be considered rather incremental: it proposes a possible improvement on an already-existing method. Still, the improvement is noticeable and relevant, at least for the test cases considered


**Questions:**

Some criticism:
- While I appreciate Lemma 1 in giving an indication of the relative size of diagonal vs non-diagonal entries, I think it requires some caveats, as I find it’s not really sufficient to justify considering only the diagonal part of the matrix. In my opinion, since you’re dropping the *whole* non-diagonal part, the ideal comparison would be between $||H^(i,i)||$ and the whole $\sum_{j\neq i} ||H^(j,j)||$; that is, ideally one should provide some indication of block-diagonal dominancy. I don’t think a bound in this form can be established, so I think it would be fair to point this out. (Mind me: I still consider the simplification valid, especially in light of the results in Fig1)
- How many different versions of Shampoo(x) have you experimented with? I’m asking because in Fig 3 it’s the one method which shows the highest variability with respect to x - and also the largest improvement (especially wrt to time) as x is decreased. It would be nice to include a Shampoo(125 / 256), to check whether this trend continues
- Why is only the train loss reported in in Table 2 and only the validation accuracy in table 3?

Some curiosity:
- Have you tried using nonlinearities other than tanh in your MLP benchmark? This is pure speculation, but I was thinking about the observation that $X$ is low-rank (you mention it in Line 149). Since we’re considering gradients, if I got everything right, this matrix should be actually further multiplied by the derivatives of the nonlinearities: I was wondering how this observation would change if different activation functions are considered (or if it would change at all!)
- I’m quite impressed that initialising $Y$ using information from the previous iteration works so well! You mention in the main paper (Line 175) that just a few iterations are necessary to convergence of the randomised SVD (and indeed you only use one iteration in your experiments), but how quickly does convergence occur? In the ablation study in the appendix (Table 5) you only report the final time/loss results, unfortunately. I ask this, because one might consider simply reusing the bases from the previous iteration, if they’re already good enough, and only update them once every few optimisation steps (as done in Shampoo? - Line 256), to further speed things up: have you investigated this?

Some minor corrections:
- I found the notation in Sec 2.1 (right before Lemma1) a bit confusing: it’s tricky to keep track of what is a vector, what is a matrix, and what is a vectorisation of a matrix. For example, $D_W$ should be a matrix (since you treat it as such in the proof in the appendix), but here you define it as a vector? Just reporting dimensions explicitly, and ensuring consistency with the notation in the appendix should help with readability
- Line 34: $g_t$ is a vector in $\mathbb{R}^{d_{in}d_{out}}$, not a matrix in $\mathbb{R}^{d_{in}\times d_{out}}$?
- In the equation right after Line 162:  I think the $+$ is a typo for $\approx$?
- In Line 179, you’re missing a transpose $M_t M_t^T$?
- In Line 406 in the appendix, you’re missing a transpose $X^TDz$?
- Appendix 6.8: is $\Omega$ actually $Y$?


**Limitations:**

A limitation of the method (in terms of memory consumption) is made explicit in the dedicated section in the paper. Still, my impression is that this could be extended (particularly, with comments pertaining the first “weakness”)

---

> ### Author Rebuttal · Authors · 2023-08-09
>
> Thank you for the detailed review!
>
> > The approach might be rather limited in its applicability, as it mostly focuses on fully-connected layers.
>
> We have experiments on the transformer, which also contains self-attention layers. In general, our method can be used to handle any tensor product operations in neural networks, which is quite universal. For convolutional layers, since each kernel (block) could be quite small, our method will become equivalent to the block-diagonal approximation (e.g., BlockAdam). The contribution of this paper is to provide a way to scale block-diagonal approximation to larger matrices using the proposed shared basis approach.
>
> > While I appreciate Lemma 1 in giving an indication of the relative size of diagonal vs non-diagonal entries, I think it requires some caveats.
>
> > ideally one should provide some indication of block-diagonal dominancy. I don’t think a bound in this form can be established, so I think it would be fair to point this out.
>
> Thank you. You are correct that we are not able to establish a bound for block-diagonal dominancy. We will mention this in the paper.
>
> > How many different versions of Shampoo(x) have you experimented with? It would be nice to include a Shampoo(125 / 256), to check whether this trend continues.
>
> In the experiment, we observed that further decreasing the block size of Shampoo does not give any speed up, so we did not further decrease the block size, as shown in the following table. The potential reason for this could be the amount of parallelism available and the implementation overhead.
>
>
>
>
> |            | Shampoo (256) | Shampoo (512) | Shampoo (1024) |
> | ---------- | ------------- | ------------- | -------------- |
> | Time (min) | 348           | 325           | 564            |
>
>
>
>
> > Why is only the train loss reported in Table 2 and only the validation accuracy in table 3?
>
> We follow the previous works such as K-fac [23] and Miniblock Fisher [5] to evaluate the train loss on the autoencoder benchmark. We have added the training loss curve for the transformer benchmark to the one page pdf for rebuttal, and our method still outperforms other optimizers in terms of training loss.
>
> > Have you tried using nonlinearities other than tanh in your MLP benchmark?
>
> Yes, the transformer benchmark uses the elu non-linearity.
>
> > You mention in the main paper (Line 175) that just a few iterations are necessary to convergence, but how quickly does convergence occur? I ask this, because one might consider simply reusing the bases from the previous iteration, if they’re already good enough, and only update them once every few optimisation steps (as done in Shampoo?)
>
> Currently we observe that the inner iteration (when setting to 1) is not the major bottleneck, so further reducing this cost may not be useful for our current experiments. However, this is a great suggestion and we will definitely try this idea later.
>
> > Some minor corrections
>
> Thank you. We have fixed these in the paper.

---

> > ### Comment · Reviewer_BzFF · 2023-08-18
> >
> > I thank the authors for addressing my concerns. In light of their comments, as well as their replies to the other reviewers, I believe the score originally given reflects the overall quality of the paper.

---

### Official Review · Reviewer_9J53 · 2023-07-21

**Soundness:** 3 good
**Presentation:** 3 good
**Contribution:** 3 good
**Rating:** 6
**Confidence:** 3

**Summary:**

Computing the full Gram matrix is practically infeasible for large models even if it's done layer-wise. This work first confirms the block structure of each layer's Gram matrix experimentally. Previous work considered approximating each of these blocks as a low-rank matrix to improve the memory and computation requirements. However, Storing the basis for each block remains problematic. Therefore the authors propose using a shared basis for all the blocks, providing an intuition why such a shared basis exists. They compare their method with previous work for training an Autoencoder on MNIST and for training a Transformer. In both cases, they show they can obtain superior or comparable results while being faster and using less memory.

**Strengths:**

The paper is written clearly and it is easy to follow and understand. The idea of using a shared basis is interesting and the authors justify the existence of such basis based on the low-rank nature of data. Furthermore, the experiments verify that the method works for training neural networks to some extent. The algorithm is also faster than Adam on the experiments towards the end.

**Weaknesses:**

Unfortunately, the evaluation of the method remains limited. The auto-encoding task is only tried with MNIST. Given that the time of the proposed method is within 4-5x of Adam, it should be feasible to also run similar experiment on larger datasets such as CIFAR10.

Another limitation of the proposed method is that it does not scale well with rank. The ablations stop at k=32 which is the rank used for the experiments. Therefore, it is not clear whether k=32 is already enough to obtain close-to-best performance or if this limitation poses a big challenge and needs to be overcome for the method to be successfully deployed in larger scales.

The theoretical arguments are also a bit limited. No convergence rate is provided for non-convex settings. Furthermore, I am not sure if the proof of Theorem 1 covers the stochastic case especially since no bound is assumed on the noise.

As a minor note, I think for the ablation studies in the appendix, using plots would work better than the tables to allow easier qualitative comparison.

**Questions:**

1. The method used for computing the shared basis is interesting. I was wondering if this method or similar ones has been used already (in which case I think it would be nice to cite them) or if this is a novel contribution? If the latter, I was wondering if there is a mathematical argument to support using this method or is it possible that some other method would work better in estimating the basis?

2. The authors mentioned that they do not pre-condition the embedding layer to be fair to other methods even though they could. What happens if the embedding layer is included? Is there a limit (though higher than 4096 for Shampoo) on the size of the embedding for your method as well? In any case I think the results would be interesting to have (in addition to the fair comparison you already performed).

**Limitations:**

The authors briefly discuss the quadratic scaling of the space complexity with rank. I think the discussion could be a bit more thorough and could also include the cubic dependence of computation complexity on the rank. Also, to make it more interpretable, it would be good to mention how slower the algorithm can be for very large models than for example Adam. For example I think using k = 32, it can be 32x slower but helps save memory especially if d[in] and d[out] are both much larger than 1024.
The authors also mention that larger models might require larger k. I think including a brief study on this in this work would make it much more clear whether the proposed method is applicable to large-scale models or not.

---

> ### Author Rebuttal · Authors · 2023-08-09
>
> Thank you for the valuable feedback.
>
> > The evaluation of the method remains limited. The auto-encoding task is only tried with MNIST. it should be feasible to also run CIFAR 10.
>
> In addition to the autoencoder benchmark, we also conduct experiments on transformers. We will further expand the autoencoder experiments in the future.
>
> > Another limitation of the proposed method is that it does not scale well with rank. The ablations stop at k=32 which is the rank used for the experiments. Therefore, it is not clear whether k=32 is already enough to obtain close-to-best performance
>
> As shown in Figure 4 of the appendix, k=32 already gives a decent approximation to the block diagonal. This is the main reason we use k=32 in our experiments. Our further experiment shows that we can use an even smaller k to obtain decent performance.
>
> For the autoencoder benchmark,
>
>
>
>
> ||Adam|Ours (rank 4)|Ours (rank 32)|Ours (rank 64)|Ours (rank 128)|
> |---|---|---|---|---|---|
> |Train Loss|54.66|52.88|51.49|51.10|50.84|
>
>
>
> For the transformer benchmark, we use the best parameters obtained from rank=32 and run for 10 runs.
>
> | Validation Accuracy | Adam  | Block Adam | Ours (rank 4) | Ours (rank 32) | Ours (rank 64) |
> | ------------------- | ----- | ---------- | ------------- | -------------- | -------------- |
> | Mean                | 67.70 | 70.36      | 70.41         | 70.51          | 70.58          |
> | Variance            | 0.16  | 0.21       | 0.03          | 0.03           | 0.04           |
>
> > Furthermore, I am not sure if the proof of Theorem 1 covers the stochastic case especially since no bound is assumed on the noise.
>
> We believe the proof of Theorem 1 covers the stochastic case, as in online convex optimization, the loss function at each step $f_t$ can be seen as a stochastic sample, which can even be chosen adversarially.
>
> > I think for the ablation studies in the appendix, using plots would work better than the tables to allow easier qualitative comparison.
>
> Thank you for the suggestion. We have added the plots to the one page pdf for rebuttal.
>
> > The method used for computing the shared basis is interesting. I was wondering if this method or similar ones has been used already.
>
> Our method can be seen as a special case of high order SVD, which guarantees quasi-optimality (the approximation error is optimal up to a factor of the square root of the dimensions, so $\sqrt{3}$ in our case). We will clarify this in the writing. The application of using high order SVD to obtain shared bases for approximating the preconditioner is our novelty.
>
> > it would be good to mention how slower the algorithm can be for very large models than for example Adam. For example I think using k = 32, it can be 32x slower but helps save memory.
>
> The time complexity for each training step can be decomposed into two parts: (1) back-propagation for gradient computation, and (2) computing the update based on gradient. For (1), all the optimizers have identical time cost. For (2), our method is faster than other second order methods (as shown in the experiments) but could be k times worse than Adam. However, in large scale training, (1) often dominates since it is proportional to the batch size. Further, for models such as transformers, the gradient computation cost goes quadratically with the sequence length. Consequently, we think our method will not be much slower than Adam for very large models. We will add more descriptions to the paper.

---

> > ### Author Response · Authors · 2023-08-18
> >
> > Dear Reviewer 9J53,
> >
> > We have addressed the weaknesses and questions pointed out in your original review. In addition, we have added the result of the autoencoder benchmark on cifar-10 (suggested by your review) as follows.
> >
> > |            | Adam    | Block Adam | Ours (rank 32) |
> > | ---------- | ------- | ---------- | -------------- |
> > | Train Loss | 1776.20 | 1764.41    | 1758.05        |
> >
> > Similar to the MNIST data set, our proposed method performs better than Adam and Block Adam.
> >
> > We also added the result on one of the gpt-2 models on WikiText-103 (over 100M tokens) with 120 trials (batch size 128, 10k steps) of random hyperparameter search as follows.
> >
> > |                       | Adam  | Ours (rank 32) |
> > | --------------------- | ----- | -------------- |
> > | Validation Perplexity | 55.92 | 54.33          |
> > | Time (min)            | 121   | 130            |
> >
> > Our method is over 25% faster in steps and 15% faster in time to reach the same level of validation perplexity as Adam.
> >
> > Please feel free to let us know if you have any additional questions. We are happy to answer any additional questions and interact with the reviewers.

---

> > > ### Comment · Reviewer_9J53 · 2023-08-19
> > >
> > > Dear Authors,
> > >
> > > Thank you very much for your replies.
> > >
> > > Regarding the discussion around time complexity: I suggest including this discussion in the next reivsion (backed by plots or references showing the dominance of backpropagation time).
> > >
> > > Regarding the effect of $k$: It seems that increasing $k$ continues to improve performance (the improvement is not plateauing). So the quadratic scaling with $k$ can indeed become a bottleneck for obtaining the best performance. It is also interesting to note that in the extreme case where $k = d_{\text{in}}$, then the algorithm has as much freedom as Block Adam with block size $d_{\text{in}}$ but still the proposed method would perform better (assuming that increasing $k$ does not hurt performance).
> > >
> > > Regarding the new results on CIFAR10 and WikiText: Thank you for running these experiments. For autoencoding, I do not have a baseline to assess whether with the obtained loss values the model can be considered converged or not. However, for WikiText, I expect GPT2 model to quickly get close to 25 perplexity. After 10k steps the perplexity should be even lower. Therefore, I suspect some parameter is not correctly chosen for these experiments.
> > >
> > > Overall, if the results are amended to reflect the state of the art performances I think they would be a great addition. While until this happens, I find the evaluation to be too limited to convince practitioners to use this method, I think the main idea and method have merits. Therefore I have updated my score to suggest acceptance but only on the borderline.

---

> > > > ### Author Response · Authors · 2023-08-21
> > > >
> > > > > However, for WikiText, I expect GPT2 model to quickly get close to 25 perplexity.
> > > >
> > > > Yes. Due to the limited time and computational resources, we currently only have the result for one of the smaller gpt2-model (7.3M params). The larger models take a lot longer to run, but we will add the results to the paper once the experiments are finished. Note that for each run we conduct hyper-parameter search over 120 hyper-parameter configurations, so running each experiment is quite costly.
> > > >
> > > > Additionally, we added the results of ResNet on Cifar10 as requested by the other reviewer as follows.
> > > >
> > > > ResNet20 (0.27M)
> > > >
> > > > |               | Adam (100k step) | Ours (100k step) | Adam (200k step) | Ours (200k step) |
> > > > | ------------- | ---------------- | ---------------- | ---------------- | ---------------- |
> > > > | Test Accuracy | 92.07            | 92.35            | 92.79            | 93.02            |
> > > >
> > > > ResNet56 (0.85M)
> > > >
> > > > |               | Adam (100k step) | Ours (100k step) | Adam (200k step) | Ours (200k step) |
> > > > | ------------- | ---------------- | ---------------- | ---------------- | ---------------- |
> > > > | Test Accuracy | 92.22            | 93.04            | 94.02            | 94.40      |
> > > >
> > > > Our method performs better than Adam on both ResNet20 and ResNet56. The performance gap naturally decreases as the number of steps increases as both methods should reach similar performance after running for a sufficient amount of steps. Furthermore, the Transformer architecture is much more “nonlinear” than ResNets, leading to more room for improvements.

---

> > > > > ### Comment · Reviewer_9J53 · 2023-08-21
> > > > >
> > > > > Dear Authors,
> > > > >
> > > > > Thank you very much for the new results. Given the demonstration of capability to reach state of the art accuracy values, I have further increased my score.
> > > > >
> > > > > Thank you very much.

---

### Official Review · Reviewer_Yba5 · 2023-07-22

**Soundness:** 4 excellent
**Presentation:** 3 good
**Contribution:** 3 good
**Rating:** 7
**Confidence:** 3

**Summary:**

Following up on the line of work stemming from AdaGrad, the paper presents a novel adaptive optimizer that uses a block-diagonal preconditioner. The block-diagonal elements of the second-moment matrix are approximated through a low-rank approximation that shares the same basis, computed via randomized SVD, for all the block-diagonal components. Regret bounds are shown to be optimal, and the performance of the optimizer is evaluated against related previous work on an autoencoder and a transformer benchmark.

**Strengths:**

The paper is on average well-written, well-presented, and technically solid. Relatively complex technical content is explained in a clear way, with adequate coverage of background knowledge.
The choices to use a block-diagonal second moment matrix and to use a shared low-rank approximation are adequately justified experimentally, and by presenting technical results in a simplified setup (under fixed parameters). Optimal regret bounds are provided, and section 2.3 appears to cover the differences with related previous work in a detailed manner.
The experimental results suggest that the proposed algorithm is more effective than more expensive methods even when plotted over optimization steps.

**Weaknesses:**

I believe the paper would benefit from a wider experimental section, perhaps including convolutional networks or vision transformers (on vision tasks). In addition, I think it would be interesting to study in more detail the regularizing effect of the various optimizers. Information concerning this could be provided by the training curves from the transformer experiment.

As a very minor comment: I think the presentation could be improved by splitting section 2 into two sections: "background" and "proposed method".

**Questions:**

- Could the authors provide the training curves for the transformer benchmark, allowing for a comparison of the generalisation properties of the different optimizers?
- Would the proposed approach pay off even on convolutional networks, for instance when compared against [5], or against Adam in terms of performance over runtime?
- How was rank=32 determined? Would performance over steps improve with larger values of $k$? The appendix only provides experiments for lower ranks.

-------------------
Update after rebuttal:

In view of the discussion of the authors with the other reviewers and the provided new experimental results, which would appear to support the validity of the proposed approach, I keep my score of 7.

**Limitations:**

Limitations are clearly detailed in the conclusions.

---

> ### Author Rebuttal · Authors · 2023-08-09
>
> Thank you for the detailed review!
>
> > I believe the paper would benefit from a wider experimental section, perhaps including convolutional networks or vision transformers (on vision tasks).
>
> We will expand the experimental section with our future experiments.
>
> > As a very minor comment: I think the presentation could be improved by splitting section 2 into two sections: "background" and "proposed method".
>
> Thanks for the suggestion! We will refine the paper following your suggestion.
>
> > Could the authors provide the training curves for the transformer benchmark, allowing for a comparison of the generalization properties of the different optimizers?
>
> Thank you for the suggestion. We have added the training curves for the transformer benchmark.
>
> > Would the proposed approach pay off even on convolutional networks, for instance when compared against [5], or against Adam in terms of performance over runtime?
>
> The Miniblock Fisher paper [5] can be seen as a special case of ours where a mini block size with full-rank shared basis is used for the convolutional layer, and the time complexity will be the same as ours. In their paper, they are outperforming Adam. We will explore in the future experiment whether different block sizes and ranks can further improve the performance.
>
>
> > How was rank=32 determined? Would performance over steps improve with larger values of $k$? The appendix only provides experiments for lower ranks.
>
> As shown in Figure 4 of the appendix, k=32 gives a decent approximation to the block diagonal. This is the main reason we use k=32 in our experiments.
>
> For the autoencoder benchmark, the performance over steps improves with larger values of $k$, as shown in the following table.
>
>
>
>
> ||Adam|Ours (rank 4)|Ours (rank 32)|Ours (rank 64)|Ours (rank 128)|
> |---|---|---|---|---|---|
> |Train Loss|54.66|52.88|51.49|51.10|50.84|
>
>
>
>
> For the transformer benchmark, the performance slightly increases, and smaller k still obtains decent performance. Therefore, we think k=32 is sufficient for both settings since it strikes a good tradeoff between performance and memory efficiency.
>
>
> | Validation Accuracy | Adam  | Block Adam | Ours (rank 4) | Ours (rank 32) | Ours (rank 64) |
> | ------------------- | ----- | ---------- | ------------- | -------------- | -------------- |
> | Mean                | 67.70 | 70.36      | 70.41         | 70.51          | 70.58          |
> | Variance            | 0.16  | 0.21       | 0.03          | 0.03           | 0.04           |

---

### Official Review · Reviewer_ssTg · 2023-07-24

**Soundness:** 2 fair
**Presentation:** 3 good
**Contribution:** 2 fair
**Rating:** 5
**Confidence:** 3

**Summary:**

This paper proposes an adaptive optimizer similar to Shampoo and GGT, which considers non-diagonal components of the preconditioner matrix. They reduce the cost of the preconditioner by using a unit-wise block diagonalization on the output units and a further low-rank approximation with a shared basis between the diagonal blocks.

**Strengths:**

The time and space complexity of preconditioned optimizers is a critical issue, especially when models keep increasing in size at a rapid pace. The proposed method strikes a good balance between accurately representing the preconditioner matrix and its time/space complexity. They compare against Adam, block Adam, Shampoo, and Sketchy on autoencoder and transformer benchmarks, and demonstrate that their

**Weaknesses:**

The novelty with respect to the existing work [4] and [11] seems incremental, and the improvement over block Adam [32] is marginal (0.2% increase in validation accuracy). The reason why the proposed method shows better validation accuracy than Shampoo is also unclear. Since the proposed method introduces further approximation to Shampoo in order to reduce the time and space complexity, any improvement in validation accuracy is certainly not by design. They choose a fixed rank k=32 for their experiments, but this number seems to be chosen arbitrarily.

**Questions:**

What is the effect of removing the smallest eigenvalue from the R matrix? The experiments don't seem to compare with and without this difference.

**Limitations:**

This paper does not have any potential negative societal impact.

---

> ### Author Rebuttal · Authors · 2023-08-09
>
> Thank you for the valuable response.
>
> > The novelty with respect to the existing work [4] and [11] seems incremental
>
>
> We are sorry that our writing might cause some confusion, but our method is very different from the work of Shampoo [4] and Sketchy [11]. Shampoo and Sketchy approximate the second-moment matrix with a Kronecker product approximation so all the diagonal and non-diagonal blocks are the same up to a scalar. In contrast, we focus on the finer approximation of the diagonal blocks – the coefficient matrices ($R_t^{(i)}$) enable the diagonal blocks to be different, while the basis sharing scheme makes the approximation memory efficient. Therefore, the proposed shared basis block low-rank approximation  is totally different from the Kronecker product approximation. We will clarify this in the paper.
>
> > The reason why the proposed method shows better validation accuracy than Shampoo is also unclear. Since the proposed method introduces further approximation to Shampoo
>
> As explained above, we are not an approximation to Shampoo. Instead, our method can approximate diagonal blocks much better than Shampoo, leading to improved performance. We will clarify this in the paper.
>
> > What is the effect of removing the smallest eigenvalue from the R matrix? The experiments don't seem to compare with and without this difference.
>
> We follow the frequent direction paper and Sketchy [11] to remove the smallest eigenvalue when applying the frequent direction method. The regret bound can be obtained with and without doing so. Our experiment on the autoencoder benchmark shows that the performances are similar.
>
>
>
>
> ||Ours (rank 32)|Ours (rank 32, without subtract)|
> |---|---|---|
> |Train Loss|51.49|51.48|

---

### Official Review · Reviewer_Z6xV · 2023-07-25

**Soundness:** 2 fair
**Presentation:** 2 fair
**Contribution:** 2 fair
**Rating:** 5
**Confidence:** 2

**Summary:**

This paper studies how to improve the memory efficiency of adaptive methods with (non-)diagonal preconditioning. The authors propose approximating the diagonal blocks of the second moment matrix by low-rank matrices and enforcing the same basis for the blocks within each layer. The authors also provide theoretical analysis and empirical studies to demonstrate the advantage of the proposed method.

----------------------------
I have read the author’s rebuttal. Although I feel that my concerns were not fully addressed, I am willing to raise my score after seeing the seriousness of the authors' response, including their careful approach to me and the other reviewers.

**Strengths:**

1. The research problem studied in this paper is practical and significant.
2. Theoretical analysis is rich.
3. Empirical studies on various datasets.

**Weaknesses:**

1. The document is not very easy to understand.
2. the proposed methodology is not clearly superior to the baseline methodology.
3. The authors did not compare the use of multiple modeling architectures

**Questions:**

I was just wondering about the performance of the proposed method in a broader dataset.

**Limitations:**

(same as Weaknesses, but looking forward to authors' response)

1. The research problem studied in this paper is practical and significant.
2. Theoretical analysis is rich.
3. Empirical studies on various datasets.

---

> ### Author Rebuttal · Authors · 2023-08-09
>
> Thank you for the valuable feedback!
>
> > The document is not very easy to understand.
>
> We are sorry for the inconvenience and will further polish the paper.
>
> > The proposed methodology is not clearly superior to the baseline methodology.
>
> Our proposed method outperforms Adam on both benchmarks. On the transformer benchmark, we gain over 3% in validation accuracy over Adam with similar time and memory consumption. Our proposed method beats other baselines with less time and memory consumption.
>
> > The authors did not compare the use of multiple modeling architectures
>
> We conduct experiments on both the autoencoder and the transformer architectures. We will add more in future experiments.

---

> ### Author Response · Authors · 2023-08-18
> **Rebuttal by Authors - 2**
>
> We have added the result on one of the gpt-2 models on WikiText-103 (over 100M tokens) with 120 trials (batch size 128, 10k steps) of random hyperparameter search as follows.
>
> |                       | Adam  | Ours (rank 32) |
> | --------------------- | ----- | -------------- |
> | Validation Perplexity | 55.92 | 54.33          |
> | Time (min)            | 121   | 130            |
>
> Our method is over 25% faster in steps and 15% faster in time to reach the same level of validation perplexity as Adam.
>
> Additionally, we also conducted an additional experiment on the autoencoder benchmark on cifar-10 data. The results are as follows.
>
> |            | Adam    | Block Adam | Ours (rank 32) |
> | ---------- | ------- | ---------- | -------------- |
> | Train Loss | 1776.20 | 1764.41    | 1758.05        |
>
> The results for ResNet20 and ResNet56 (the small ones on CIFAR10) are as follows. We conduct 60 trials of random search for each setting.
>
>
> ResNet20 (0.27M)
>
> |               | Adam (100k step) | Ours (100k step) | Adam (200k step) | Ours (200k step) |
> | ------------- | ---------------- | ---------------- | ---------------- | ---------------- |
> | Test Accuracy | 92.07            | 92.35            | 92.79            | 93.02            |
>
> ResNet56 (0.85M)
>
> |               | Adam (100k step) | Ours (100k step) | Adam (200k step) | Ours (200k step) |
> | ------------- | ---------------- | ---------------- | ---------------- | ---------------- |
> | Test Accuracy | 92.22            | 93.04            | 94.02            | 94.40      |
>
> Our method performs better than Adam on both ResNet20 and ResNet56. The performance gap naturally decreases as the number of steps increases as both methods should reach similar performance after running for a sufficient amount of steps. Furthermore, the Transformer architecture is much more “nonlinear” than ResNets, leading to more room for improvements.

---

> > ### Comment · Reviewer_Z6xV · 2023-08-21
> >
> > I appreciate the authors' response, especially for providing new experimental results, the gpt-2 models on WikiText-103, to support their claim.
> >
> > Although I feel that my concerns were not fully addressed, I am willing to raise my score after seeing the seriousness of the authors' response, including their careful approach to me and the other reviewers. I would like to conclude by saying that our questions or concerns are not meant to be an attack on you, but rather where we would like this work to be further improved. I hope the authors can take our reviewers' comments seriously.

---

### Official Review · Reviewer_auuW · 2023-07-25

**Soundness:** 3 good
**Presentation:** 3 good
**Contribution:** 3 good
**Rating:** 6
**Confidence:** 5

**Summary:**

This paper implements an online block version of AdaGrad to capture off-diagonal information in the preconditioner at each block level. A low-rank approximation is used for each block, which is a common approach in the literature to decrease time and memory complexities. To further reduce the memory usage, a shared basis approach across all blocks is proposed, which is justified both theoretically and practically. The proposed optimizer is tested on two benchmarks which are used in previous works.

**Strengths:**

- the paper provides convergence analysis in the Online Convex Optimization framework, which might serve as a good starting point for methods that would build on top of this paper
- the idea of sharing the basis across all blocks is justified theoretically and practically (visually) in section 2.

**Weaknesses:**

- as stated in the limitations section, the memory complexity scales quadratically with rank k, which might be partially solved by decreasing k. However, since the basis is shared, there might be a lower bound on k
- the results do not have confidence intervals and the paper does not state how many random seeds were used. This is mandatory especially when the results for each method have similar values
- in the experiments, Sketchy had rank 256, which is much larger compared to k=32 for your shared-basis approach. The reviewer believes that this is not a fair comparison and this could be fixed with some additional experiments to cover these issues

**Questions:**

- how does the proposed optimizer behave with smaller ranks  $k \in \{1, 2, 4, 8 \}$? Would it be possible to provide some ablation study for this?
- in the appendix it is stated that the hyperparameter search was also performed for $1-\beta_2$, which seems to be a second order momentum parameter. How is this applied to the proposed optimizer? Does $\beta_2$ decay the outer product of the gradient $g_t$ or is it employed in the momentum applied to the update, such as $v_t^{(i)} = \beta_2 v_{t-1}^{(i)} + (1-\beta_2) u_t^{(i)}$?
- would it be possible to run Sketchy and the proposed optimizer with the same rank to have a fair comparison against them? The reviewer is aware of the relationship between the rank and block size for Sketchy and believes that an ablation study on the block size of it with fixed rank (for Sketchy and the proposed work) should have been performed. (The most important thing is to have similar ranks to be able to compare memory and, of course, block size also matters for Sketchy)
- can you compare against Adam at least on small BERT models (tiny with 4M params and/or mini with 11M params or even larger variants if time and memory allows) on GLUE-MNLI and SQuAD v1/v2? These are baseline tasks and models for NLP and benchmarking against them would be important. Moreover, in the manuscript at lines 290-292 it is stated that the proposed optimizer can handle large embedding layers and this would be a good test.
- do the learning rate scheduler and weight decay / L2-regularization (coupled or decoupled) affect the results in any way?
- based on the formula for the update provided at line 190, the final expression can be rewritten as $u_t^{(i)} = -B_t \left(R_t^{(i)} + (\rho_t^{(i)} + \epsilon)I\right)^{-1/2}B_t^T g_t^{(i)} - (\rho_t^{(i)} + \epsilon)^{-1/2}(I - B_t B_t^T) g_t^{(i)}$. Can you please explain why the term $I - B_t B_t^T$ appears in the final expression of $u_t^{(i)}$?
- if we assume that the shared base $B_t$ is orthogonal, e.g. $B_t B_t^T = I$, then working out the first expression for the update would result in $u_t^{(i)} = B_t \left(R_t^{(i)} + (\rho_t^{(i)} + \epsilon)I\right)^{-1/2}B_t^T g_t^{(i)}$. However, this is different from the update in formula $(1)$ and the reviewer believes that the update should have been $B_t \left[(R_t^{(i)} + \rho_t^{(i)})^{1/2} + \epsilon I\right]^{-1} B_t^T g_t^{(i)}$. It might be possible that the reviewer misses something, can you please clarify why the update $(1)$ is implemented differently?

Since the reviewer cares about the correctness of the formulas, some potential typos that do not influence the decision are highlighted:
- at the formula of $A_t$ between lines 167 and 168 in the manuscript, the reviewer believes that a transposition operator is missing, even though the matrices $M_t^{(i)}$ are symmetric (this is not stated in the paper). As a consequence, it might also be needed in the formulas between lines 179 and 180 (here, the index in the first sum goes up to $d/b$ instead of $m$, which is essentially the same thing).

**Limitations:**

The paper has a brief section about limitations. The reviewer would like to add that the lower memory usage and lower training time would have a positive impact in reducing power usage and carbon emissions.

---

> ### Author Rebuttal · Authors · 2023-08-09
>
> Thank you for the detailed review!
>
> > the memory complexity scales quadratically with rank k.
>
> As stated in the paper, the memory complexity scales quadratically with the rank k. But since each block has a different coefficient matrix, our approach requires a much smaller k compared with Sketchy. As shown in the experiments, our method with k=32 outperforms Sketchy with k=256.
>
> Additionally, as shown in Figure 4 of the appendix, k=32 gives a decent approximation to the block diagonal. This is the main reason we use k=32 in our experiments. Our further experiment below shows that we can use an even smaller k to obtain decent performance.
>
> > the results do not have confidence intervals and the paper does not state how many random seeds were used
>
> Thank you for the advice. We added 10 runs of the best hyperparameter on the transformer benchmark and the results are as follows.
>
>
> |Validation Accuracy |Adam|Block Adam|Ours (rank 32)
> | --- | --- | --- | --- |
> |Mean|67.70|70.36|70.51|
> |Variance|0.16|0.21|0.03|
>
>
>
> > how does the proposed optimizer behave with smaller ranks
>
> For the autoencoder benchmark,
>
>
>
> ||Adam|Ours (rank 1)|Ours (rank 2)|Ours (rank 4)|Ours (rank 8)|Ours (rank 16)|Ours (rank 32)|
> |---|---|---|---|---|---|---|---|
> |Train Loss|54.66|55.29|53.51|52.88|52.42|51.94|51.49|
>
>
> For the transformer benchmark, we use the best parameters obtained from rank=32, and run it with rank=1 and rank=4 for 10 runs.
>
> |Validation Accuracy |Adam|Block Adam|Ours (rank 1)|Ours (rank 4)|Ours (rank 32)|
> |---|---|---|---|---|---|
> |Mean|67.70|70.36|70.30|70.41|70.51|
> |Variance|0.16|0.21|0.04|0.03|0.03|
>
> The results suggest that decreasing the k will slightly hurt the performance. Further, the drop on the validation error (for transformer training) is small when decreasing k.
>
>
> > Does $\beta_2$ decay the outer product of the gradient $g_t$
>
> Yes, it decays the outer product of the gradient $g_t$. We will clarify this in the revision.
>
>
> > would it be possible to run Sketchy and the proposed optimizer with the same rank to have a fair comparison against them?
>
> We added the following experiment to compare our method with Sketchy under the same rank.
>
> For the autoencoder benchmark,
>
>
>
> |            | Sketchy (block 250, rank 32) | Sketchy (block 1000, rank 32) | Sketchy (block 1000, rank 256) | Ours (rank 32) |
> | ---------- | ---------------------------- | ----------------------------- | ------------------------------ | -------------- |
> | Train Loss | 52.73                        | 52.76                         | 52.41                          | 51.49          |
>
>
>
> For the transformer benchmark, we use the best parameters obtained from Sketchy rank=128 to run rank=32 for 10 runs.
>
>
> | Validation Accuracy | Sketchy (block 1024, rank 32) | Ours (rank 32) |
> | ------------------- | -------------------------------- | -------------- |
> | Mean                | 67.52                            | 70.51          |
> | Variance            | 0.32                             | 0.03           |
>
> The results suggest that our method outperforms Sketchy under all these settings.
>
> > do the learning rate scheduler and weight decay / L2-regularization (coupled or decoupled) affect the results in any way?
>
> We use the default learning rate scheduler in both benchmarks. The auto-encoder benchmark comes with a linear decay and the Transformer benchmark comes with a square root decay. We outperform the other methods on both benchmarks, showing that our method is not sensitive to the learning rate scheduler.
>
> > based on the formula for the update provided at line 190, the final expression can be rewritten as $u_t^{(i)}=-B_t \left( R_t^{(i)}+(\rho_t^{(i)}+\epsilon)I\right)^{-1/2}B_t^Tg_t^{(i)}-(\rho_t^{(i)}+\epsilon)^{-1/2}(I-B_tB_t^T)g_t^{(i)}$
>
> You are correct that the final expression can be rewritten as above. The term $I-B_tB_t^T$ appears in the final expression of $u_t^{(i)}$ as the projection matrix to the orthogonal space of the shared basis. So the two parts of the update can be seen as the update in the shared basis space (involving both $R_t$ and $\rho_t^{(i)}+\epsilon$) and the update in the orthogonal space (only involving $\rho_t^{(i)}+\epsilon$).
>
> > However, this is different from the update in formula $(1)$
>
> Thanks for pointing this out! Indeed this expression is different from the update in formula $(1)$. The difference between the two is adding $\epsilon$ before or after the matrix square root. The Adagrad paper uses the latter, while we follow Shampoo to use the former in our implementation. Since both can be implemented efficiently, we will fix this in the writing and compare them in future experiments.
>
> > Since the reviewer cares about the correctness of the formulas, some potential typos that do not influence the decision are highlighted
>
> Thanks! We have fixed them in our writing.

---

> > ### Comment · Reviewer_auuW · 2023-08-13
> > **Follow up on authors' rebuttal**
> >
> > I would like to thank authors for addressing most of the issues that I pointed out. Here are my comments about the rebuttal and I would like to ask the authors to read my answer carefully and exclusively pay attention to **Question (4)**.
> >
> > **Weakness (1) Lower Bound on k** Indeed, Figure 4 shows the Frobenius norm of the approximation when varying k and I was curious about the performance of the algorithm when using k smaller than 32.
> >
> > **Weakness (2) Confidence intervals** I appreciate that you added 10 different runs for the experiments. It seems like with rank 2 your approach already outperform Adam.
> >
> > **Weakness (3) Sketchy rank 256 VS your rank=32** Related to Question (3) below.
> >
> > **Question (1) Ablation on k** I am positively impressed that training does not diverge for small values of k. A few more comments about your tables that do ablation on k.
> >
> > *Autoencoder benchmark.* How do you choose the hyper-parameters for the autoencoder benchmark? Do you apply the same approach as for the transformers benchmark?
> >
> > *Transformer benchmark.* In your results from the rebuttal I notice that the results are similar for this task as for the Block Adam. Can you please report the memory usage for Adam, Block Adam and your approach using ranks 1, 2, 4, 8, 16? Also, is there any particular reason why this benchmark doesn't have results for ranks 8 and 16?
> >
> > **Question (2) About $\beta_2$** Can you please briefly explain here how $\beta_2$ decays the outer product before modifying the manuscript?
> >
> > **Question (3) Ablation study for Sketchy rank** I am satisfied with the response for this point.
> >
> > **Question (4) Comparison Against Adam on BERT** This question has been skipped without even being mentioned at all. Is there a particular reason for why you are only sticking to the Universal Dataset benchmark? This seems to be a bit suspect to me because GLUE and SquadV2 are state of the art benchmarking datasets. In addition to this, would it be possible to provide some more results on classification tasks for some ResNet models at least on CIFAR-10 or CIFAR-100? MNIST and Universal Dependencies seem to be toy datasets for the models that you used in your evaluation.
> >
> > **Question (5) About regularization** I am satisfied with the response for this point.
> >
> > **Question (6)** Does this extra term related to the projection matrix to the orthogonal space of the shared basis appear naturally from a different update or is it added artificially to improve the update direction? In case it arises naturally, can you please provide the derivation here? I am really interested in this.
> >
> > **Question (7)** I am satisfied with the response for this question. Please clarify this part in the manuscript because it is very important where dampening is applied.
> >
> > **Question (8)** Thank you for fixing these small typos in the manuscript.
> >
> > Some minor comment: I would have appreciated a bit more structure in your answers, but that's not a big issue and it doesn't affect the final decision for the paper.

---

> > > ### Author Response · Authors · 2023-08-18
> > >
> > > We thank the reviewer for the response.
> > >
> > > ---
> > >
> > > > **Question (1) Ablation on k** *Autoencoder benchmark.* How do you choose the hyper-parameters for the autoencoder benchmark?
> > >
> > >
> > > For the autoencoder benchmark, we conducted a full hyperparameter search for each k since it runs much faster than the transformer benchmark. The result of reusing the best parameters obtained from rank=32 will be as follows, which shows the same trend.
> > >
> > > |            | Adam  | Ours (rank 1) | Ours (rank 2) | Ours (rank 4) | Ours (rank 8) | Ours (rank 16) | Ours (rank 32) |
> > > | ---------- | ----- | ------------- | ------------- | ------------- | ------------- | -------------- | -------------- |
> > > | Train Loss | 54.66 | 60.52| 55.52         | 53.68         | 52.71         | 51.96          | 51.49          |
> > >
> > > > *Transformer benchmark.* Can you please report the memory usage for Adam, Block Adam and your approach using ranks 1, 2, 4, 8, 16? Also, is there any particular reason why this benchmark doesn't have results for ranks 8 and 16?
> > >
> > > The memory usage of each method is as follows. We just wanted to show the trend so didn’t include the results for rank=8, 16. The full results are presented in the following table.
> > >
> > > |                     | Adam  | BlockAdam | Ours (rank 1) | Ours (rank 4) | Ours (rank 8) | Ours (rank 16) | Ours (rank 32) |
> > > | ------------------- | ----- | --------- | ------------- | ------------- | ------------- | -------------- | -------------- |
> > > | Validation Accuracy | 67.70 | 70.36     | 70.30         | 70.41         | 70.35         | 70.38          | 70.51          |
> > > | Time (min)          | 123   | 193       | 140           | 143           | 153           | 149            | 161            |
> > > | Memory (MB)         | 19157 | 21598     | 19550         | 19553         | 19553         | 19553          | 19553          |
> > >
> > > Our method not only performs better than BlockAdam but also runs faster and takes less memory. The difference in performance, time, and space is even more significant on the autoencoder benchmark. Additionally, Figure 4 in the appendix shows that our shared basis approximation is a lot better than the block diagonal approximation.
> > >
> > > ---
> > >
> > > > **Question (2)** Can you please briefly explain here how $\beta_2$ decays the outer product before modifying the manuscript?
> > >
> > > To decay the outer product of $g_t$ with $\beta_2$,
> > > we change the equation after line 162 from
> > >
> > > $$M_t^{(i)}=B_{t-1}R_{t-1}^{(i)}(B_{t-1})^T+g_t^{(i)}(g_t^{(i)})^T$$
> > >
> > > to
> > >
> > > $$M_t^{(i)}=\beta_2 B_{t-1}R_{t-1}^{(i)}(B_{t-1})^T+(1-\beta_2)g_t^{(i)}(g_t^{(i)})^T$$
> > >
> > > and the rest just follows the original procedure.
> > >
> > > ---
> > >
> > > > **Question (4) Comparison Against Adam on BERT**
> > >
> > > We are sorry for not mentioning this in the rebuttal. Since several reviewers asked for additional experiments, we decided to run a large-scale experiment on training the GPT-2 models, but the experiments were not finished before the rebuttal deadline.
> > >
> > > The result on one of the gpt-2 models on WikiText-103 (over 100M tokens) with 120 trials (batch size 128, 10k steps) of random hyperparameter search is as follows.
> > >
> > > |                       | Adam  | Ours (rank 32) |
> > > | --------------------- | ----- | -------------- |
> > > | Validation Perplexity | 55.92 | 54.33          |
> > > | Time (min)            | 121   | 130            |
> > >
> > > Our method is over 25% faster in steps and 15% faster in time to reach the same level of validation perplexity as Adam.
> > >
> > > Additionally, we also conducted an additional experiment on the autoencoder benchmark on cifar-10 data. The results are as follows.
> > >
> > > |            | Adam    | Block Adam | Ours (rank 32) |
> > > | ---------- | ------- | ---------- | -------------- |
> > > | Train Loss | 1776.20 | 1764.41    | 1758.05        |
> > >
> > > Similar to the MNIST data set, our proposed method performs better than Adam and Block Adam.
> > >
> > > ---
> > >
> > > > **Question (6)** Does this extra term related to the projection matrix to the orthogonal space of the shared basis appear naturally from a different update or is it added artificially to improve the update direction?
> > >
> > > Yes, it appears naturally. For simplicity, let
> > >
> > > $$c=(\rho_t^{(i)}+\epsilon).$$
> > >
> > > Since $B_t^TB_t=I$, we have
> > >
> > > $$B_t^T(I-B_tB_t^T)=B_t^T-B_t^T=0$$
> > >
> > > and
> > >
> > > $$(I-B_tB_t^T)B_t=B_t-B_t=0.$$
> > >
> > > Consequently, we have
> > >
> > > $$[B_t  (R_t^{(i)}+cI)^{-1/2}B_t^T+c^{-1/2}(I-B_tB_t^T)]^2=[B_t  (R_t^{(i)}+cI)^{-1}B_t^T+c^{-1}(I-B_tB_t^T)],$$
> > >
> > > as the other two terms are $0$.
> > >
> > > Similarly, we have
> > >
> > > $$[B_t  (R_t^{(i)}+cI)^{-1}B_t^T+c^{-1}(I-B_tB_t^T)][B_t  (R_t^{(i)}+cI)B_t^T+c(I-B_tB_t^T)]=B_tB_t^T+(I-B_tB_t^T)=I.$$
> > >
> > > Combining the two equations proves that
> > >
> > > $$[B_t  (R_t^{(i)}+cI)^{-1/2}B_t^T+c^{-1/2}(I-B_tB_t^T)] = [B_t  (R_t^{(i)}+cI)B_t^T+c(I-B_tB_t^T)]^{-1/2}=[B_t  (R_t^{(i)})B_t^T+cI]^{-1/2}.$$

---

> > > > ### Comment · Reviewer_auuW · 2023-08-18
> > > > **Follow up on authors' rebuttal #2**
> > > >
> > > > I would like to thank the authors for answering most of my comments.
> > > >
> > > > I appreciate that you added the GPT-2 results, but I still think that this additional experiment does not provide a complete view over how the optimizer behaves for a wide range of tasks. The request for image classification task on ResNet (which is also an important area) was still omitted. It would have been great to see some image classification benchmarks besides autoencoder and GPT.
> > > >
> > > > I believe that there was still enough time for these experiments and I would really appreciate if you could add some experiments with ResNets-20 (the small one for CIFAR) and/or ResNet-18 (the larger one with 11.7M params) on CIFAR-10 and/or CIFAR-100 because these results could be added in the final version and would make the experimental section cover almost all areas of interest.
> > > >
> > > > Please take my comments as constructive because I want you to improve the paper (I totally understand that sometimes authors struggle with experiments on the last mile and there might not be enough time to cover everything, but please do not ignore the potential improvement suggestions, expecially when the reviewers are open and communicative).

---

> > > > > ### Author Response · Authors · 2023-08-21
> > > > >
> > > > > Thank you for the response. We definitely take your comments as constructive and we are trying our best to add more experiments. For all the experiments, we conduct a thorough hyper-parameter search for each method, therefore requiring quite a substantial amount of computational resources for every experiment.
> > > > >
> > > > > The results for ResNet20 and ResNet56 (the small ones on CIFAR10) are as follows. We conduct 60 trials of random search for each setting.
> > > > >
> > > > > ResNet20 (0.27M)
> > > > >
> > > > > |               | Adam (100k step) | Ours (100k step) | Adam (200k step) | Ours (200k step) |
> > > > > | ------------- | ---------------- | ---------------- | ---------------- | ---------------- |
> > > > > | Test Accuracy | 92.07            | 92.35            | 92.79            | 93.02            |
> > > > >
> > > > > ResNet56 (0.85M)
> > > > >
> > > > > |               | Adam (100k step) | Ours (100k step) | Adam (200k step) | Ours (200k step) |
> > > > > | ------------- | ---------------- | ---------------- | ---------------- | ---------------- |
> > > > > | Test Accuracy | 92.22            | 93.04            | 94.02            | 94.40      |
> > > > >
> > > > > Our method performs better than Adam on both ResNet20 and ResNet56. The performance gap naturally decreases as the number of steps increases as both methods should reach similar performance after running for a sufficient amount of steps. Furthermore, the Transformer architecture is much more “nonlinear” than ResNets, leading to more room for improvements.

---

> > > > > > ### Comment · Reviewer_auuW · 2023-08-21
> > > > > > **Final comment for authors' rebuttal**
> > > > > >
> > > > > > Dear Authors,
> > > > > >
> > > > > > Thank you for answering all my questions. I appreciate you added experiments for computer vision and I am updating my score from 5 to 6 because the improvements are not significant in terms of memory savings, running times or performance. The idea is interesting, but not completely new. I would like authors to take my comments as constructive.
> > > > > >
> > > > > > Thank you,
> > > > > > Reviewer auuW

---

### Author Rebuttal · Authors · 2023-08-09

We thank all the reviewers for the valuable feedback. We have added figures for the ablation study and the training curves for the transformer benchmark.

---

### Decision · Program_Chairs · 2023-09-21

**Decision:**

Accept (poster)

**Comment:**

The submission presented a block-diagonal version of AdaGrad, using a low-rank approximation for each block and a shared basis across blocks; this massively reduces memory complexity, and makes time cost manageable. The authors show good results across standard benchmarks.

The authors and reviewers were active during the discussion, and the authors provided significant additional experimental data which led the reviewers and AC to converge to acceptance.

I would encourage the authors to incorporate the ample feedback received during the discussion period in the next version of their work.